# Backdoor Mitigation by Correcting Activation Distribution Alteration

## Abstract

Backdoor (Trojan) attacks are an important type of adversarial exploit against deep neural networks (DNNs), wherein a test instance is (mis)classified to the attacker's target class whenever a backdoor trigger is present. In this paper, we reveal and analyze an important property of backdoor attacks: a successful attack causes an alteration in the distribution of internal layer activations for backdoor-trigger instances, compared to that for clean instances. Even more importantly, we find that instances with the backdoor trigger will be correctly classified to their original source classes if this distribution alteration is reversed. Based on our observations, we propose an efficient and effective method that achieves post-training backdoor mitigation by correcting the distribution alteration using reverse-engineered triggers. Notably, our method does not change *any* trainable parameters of the DNN, but achieves generally better mitigation performance than existing methods that do require intensive DNN parameter tuning. It also efficiently detects test instances with the trigger, which may help to catch adversarial entities.

## 1 Introduction

Deep neural networks (DNN) have shown impressive performance in many applications, but are vulnerable to adversarial attacks. Recently, backdoor (Trojan) attacks have been proposed against DNNs used for image classification (Gu et al. (2019); Chen et al. (2017); Nguyen & Tran (2021); Li et al. (2019); Saha et al. (2020); Li et al. (2021a)), speech recognition (Liu et al. (2018b)), text classification (Dai et al. (2019)), point cloud classification (Xiang et al. (2021)), and even deep regression (Li et al. (2021b)). The attacked DNN will classify to the attacker's target class whenever a test instance is embedded with the attacker's backdoor trigger, while maintaining high accuracy on backdoor-free instances. Typically, a backdoor attack is launched by poisoning the training set of the DNN with a few instances embedded with the trigger and (mis)labeled to the target class.

Most existing works on backdoors either focus on improving the stealthiness of attacks (Zhao et al. (2022); Wang et al. (2022b)), their flexibility for launching (Bai et al. (2022); Qi et al. (2022)), their adaptation for different learning paradigms (Xie et al. (2020); Yao et al. (2019); Wang et al. (2021)), or develop defenses for different practical scenarios (Du et al. (2020); Liu et al. (2019); Dong et al. (2021); Chou et al. (2020); Gao et al. (2019)). However, there are few works studying the basic properties of backdoor attacks. Tran et al. (2018) first observed that triggered instances (labeled to the target class) are separable from clean target class instances in terms of internal layer activations of the poisoned classifier. This property led to defenses that detect and remove triggered instances from the poisoned training set (Chen et al. (2019a); Xiang et al. (2019)). As another example, Zhang et al. (2022) studied the differences between the parameters of clean and attacked classifiers, which inspired a stealthier attack with minimum degradation in accuracy on clean test instances.

In this paper, we investigate an interesting *distribution alteration* property of backdoor attacks. In short, the learned backdoor trigger causes a change in the distribution of internal activations for test instances with the trigger, compared to that for backdoor-free instances; and we demonstrate that instances with the trigger are classified to their original source class after the distribution alteration is reversed. Accordingly, we propose a method to mitigate backdoor attacks (post-training), such that classification accuracy on instances both with and without the trigger will be close to the accuracy of a clean (backdoor-free) classifier. In particular, we propose a practical way to correct the distribution alteration by exploiting reverse-engineered triggers (Wang et al. (2019); Xiang et al. (2020)). Compared with existing approaches that address the same mitigation problem, but which require tuning of the whole DNN, our method achieves generally better performance and without changing

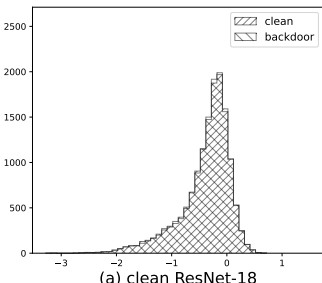 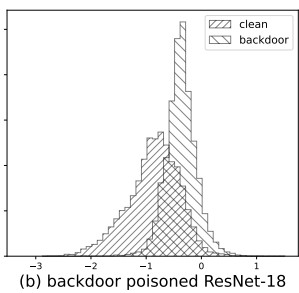 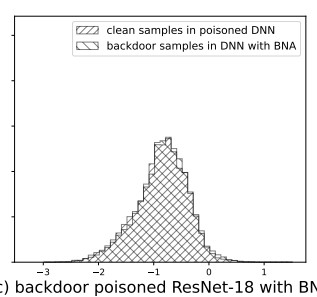

(a) clean ResNet-18     (b) backdoor poisoned ResNet-18     (c) backdoor poisoned ResNet-18 with BNA

Figure 1: Activation distribution of a neuron in the penultimate layer of ResNet-18 trained on CIFAR-10, for instances with and without a backdoor trigger, for (a) a clean classifier and (b) a backdoor-poisoned classifier (with the same trigger). In (c), the distribution alteration in (b) is reversed by our proposed method – most instances with the trigger will thus be correctly classified.

*any* original parameters of the DNN. Moreover, while most mitigation approaches are designed to correctly classify backdoor-trigger instances blindly without detection, our method is able to detect those backdoor-trigger instances efficiently. Our main contributions in this paper are twofold:

1) We discover and analyze the activation distribution alteration property of backdoor attacks and its relation to accuracy in classifying backdoor-triggered instances.

2) We propose a post-training backdoor mitigation approach based on our findings, which outperforms several state-of-the-art approaches for a variety of datasets and backdoor attack settings.

## 2 RELATED WORK

Existing backdoor defenses are deployed either during the DNN's training stage or post-training. The ultimate goal of training-stage defenses is to train an accurate, backdoor-free DNN given the possibly poisoned training set. To achieve this goal, Shen & Sanghavi (2019); Huang et al. (2022); Li et al. (2021d); Chen et al. (2019a); Xiang et al. (2019); Du et al. (2020) either identify a subset of "high-credible" instances for training, or detect and then remove training instances possibly with a backdoor trigger before training. Post-training defenders, however, are assumed to have no access to the classifier's training set. Many post-training defenses aim to detect whether a given classifier has been backdoor-compromised. Wang et al. (2019); Xiang et al. (2020); Wang et al. (2020); Liu et al. (2019) perform anomaly detection using triggers reverse-engineered on an assumed independent clean dataset; while Xu et al. (2021); Kolouri et al. (2020) train a (binary) meta classifier on "shadow" classifiers trained with and without attack.

However, model-detection defenses are not able to mitigate backdoor attacks at test time. Thus, there is a family of post-training backdoor mitigation approaches proposed to fine-tune the classifier on the assumed clean dataset, with a subset of neurons possibly associated with the backdoor attack pruned (Liu et al. (2018a); Wu & Wang (2021); Guan et al. (2022); Zheng et al. (2022)), by leveraging knowledge distillation to preserve only classification functions for clean instances (Li et al. (2021c); Xia et al. (2022)), or by solving a min-max problem as an analogue to adversarial training for evasion attacks (Zeng et al. (2022); Madry et al. (2018)). These methods all aim to enhance the robustness of the classifier against triggers embedded at test time, but are not implemented with a backdoor detector. The cost of such robustness is usually a significant degradation in the classifier's accuracy on clean instances, especially when the clean data for fine-tuning are insufficient. Another family of approaches are designed to detect test instances embedded with the trigger, without altering the classifier (Gao et al. (2019); Chou et al. (2020); Doan et al. (2020)). Defenses in this category may help to catch the adversarial entities in the act, but they cannot correctly classify the detected backdoor trigger instances to their original source classes. Moreover, existing methods in this category require heavy computation at test time (where rapid inferences are needed). In contrast, our mitigation framework includes both test-time trigger detection and source class inference, both with very little computation, as will be detailed in Sec. 4.2.

Closely related to our method, Neural Cleanse (NC) proposed by Wang et al. (2019) detects backdoor attacks and then fine-tunes the classifier using a reverse-engineered trigger. However, NC is not as effective as our method in backdoor mitigation, especially when its fine-tuning is performed with insufficient data (see the last paragraph in Sec. 5.2 for more details). Moreover, NC does not detect backdoor-trigger instances during inference, unlike our method.

# 3 DISTRIBUTION ALTERATION PROPERTY OF BACKDOOR ATTACKS

In this section, we first present the activation distribution alteration property of backdoor attacks. Then for a simplified setting, we analytically show how closing the "gap" between the clean-instance and backdoor-trigger instance distributions improves the accuracy in classifying backdoor-trigger instances; this will guide the design of our backdoor mitigation approach in Sec. 4.

**Property 3.1. (Activation Distribution Alteration)** *For a successful backdoor attack, two different backdoor-trigger instances will induce perturbations to the activations of an internal DNN layer that are in a similar direction. Thus, there is effectively a "shift" in the internal layer activation distribution for backdoor-trigger instances, compared to that for backdoor-free instances.*

Distribution alteration can be easily visualized empirically. Consider a set of clean instances from CIFAR-10 (Krizhevsky & Hinton (2009)) and the *same* set of instances but with the backdoor trigger used by Gu et al. (2019) embedded in each instance. For a ResNet-18 (He et al. (2016)) classifier that was successfully attacked using this trigger, there is a *divergence* between the distributions of the internal layer activations induced by these two sets of instances. This is shown in Fig. 1b for a neuron in the penultimate layer as an example. In comparison, for a clean classifier (not backdoor-attacked), the divergence between the two distributions is almost negligible as shown in Fig. 1a. Based on these visualizations, we ask the following question: Suppose the distribution alteration is *reversed* for each neuron, e.g. by applying a transformation to the internal activations of the triggered instances, so that the transformed distribution now closely agrees with the distribution for clean (without the backdoor-trigger) instances (see Fig. 1c). Then, following this compensation, will the classifier accurately predict the true class of origin for these backdoor-trigger instances?

Here, we investigate this problem in a simplified binary classification setting similar to the one considered by Ilyas et al. (2019). For a clean training random vector $(\mathbf{X}, Y)$ with a uniform class prior, i.e. $Y \sim \mathcal{U}\{-1, +1\}$ and with $\mathbf{X}|Y \sim \mathcal{N}(Y \cdot \boldsymbol{\mu}, \Sigma)$, where $\boldsymbol{\mu} \in \mathbb{R}^d$ and $\Sigma = \sigma^2 \boldsymbol{I}$, consider a backdoor attack with *target class* '+1', *triggered instance* $\mathbf{X}_b \sim \mathcal{N}(\boldsymbol{\mu}_b, \Sigma_b)$ with $\boldsymbol{\mu}_b = -\boldsymbol{\mu} + \boldsymbol{\epsilon}$, and $\Sigma_b = \sigma_b^2 \boldsymbol{I}$. Here, class '−1' is automatically the *source class* of $\mathbf{X}_b$ since there are only two classes. With backdoor poisoning, a multi-layer perceptron (MLP) classifier is trained with one hidden layer of $J$ nodes, a batch normalization (BN) layer (Ioffe & Szegedy (2015)) followed by linear activation, and two output nodes with functions $f_- : \mathbb{R}^d \rightarrow \mathbb{R}$ and $f_+ : \mathbb{R}^d \rightarrow \mathbb{R}$ corresponding to classes '−1' and '+1' respectively. An instance $\boldsymbol{x}$ will be classified to class '−1' if $f_-(\boldsymbol{x}) > f_+(\boldsymbol{x})$; else it will be classified to '+1'.

**Definition 3.1. ($\eta$-erroneous classifier)** *A classifier is said to be $\eta$-erroneous if the error rate for each class is upper bounded by $\eta$.*

**Definition 3.2. ($\psi$-successful attack)** *A backdoor attack is said to be $\psi$-successful if its attack success rate (ASR), i.e. the probability for triggered instances being (mis)classified to the attacker's target class (Li et al. (2022b)), is at least $\psi$; in our case, this means that $P[f_+(\mathbf{X}_b) > f_-(\mathbf{X}_b)] \geq \psi$.*

Given the settings above, for an arbitrary input $\boldsymbol{x}$, the activation of the $j$-th node ($j \in \{1, \cdots, J\}$) (after BN with trained parameters $\gamma_j$ and $\beta_j$), with weight vector $\boldsymbol{w}_j$ in the hidden layer, is:

$$a_j(\boldsymbol{x}) = \frac{\boldsymbol{w}_j^\top \boldsymbol{x} - m_j}{\sqrt{v_j}} \gamma_j + \beta_j, \tag{1}$$

where $m_j$ and $v_j$ respectively are the mean and variance stored by the BN layer during training on the *poisoned training set*. Then the activation distribution for clean source class instances $(\mathbf{X}|Y = -1) \sim \mathcal{N}(-\boldsymbol{\mu}, \Sigma)$ is a Gaussian specified by mean $\mathbb{E}[a_j(\mathbf{X})|Y = -1]$ and variance $\text{Var}[a_j(\mathbf{X})|Y = -1]$; while for triggered instances $\mathbf{X}_b \sim \mathcal{N}(\boldsymbol{\mu}_b, \Sigma_b)$, the activation follows a Gaussian specified by mean $\mathbb{E}[a_j(\mathbf{X}_b)]$ and variance $\text{Var}[a_j(\mathbf{X}_b)]$. An easy way to eliminate the *divergence* between these two distributions is to create a classifier *for triggered instances* $\mathbf{X}_b$[1] by replacing $a_j$ in Eq. (1) with $a_j^*(\boldsymbol{x}) = (\boldsymbol{w}_j^\top \boldsymbol{x} - m_j^*)\gamma_j / \sqrt{v_j^*} + \beta_j$ for each node $j$, where (see Apdx. A.1 for derivation):

$$m_j^* = \frac{\sigma_b}{\sigma} m_j + (\frac{\sigma_b}{\sigma} - 1)\boldsymbol{w}_j^\top \boldsymbol{\mu} + \boldsymbol{w}_j^\top \boldsymbol{\epsilon} \quad \text{and} \quad v_j^* = \frac{\sigma_b}{\sigma} v_j, \tag{2}$$

---

[1]These can be constructed in practice, given an estimated backdoor trigger, by embedding the trigger in clean instances available to the defender.

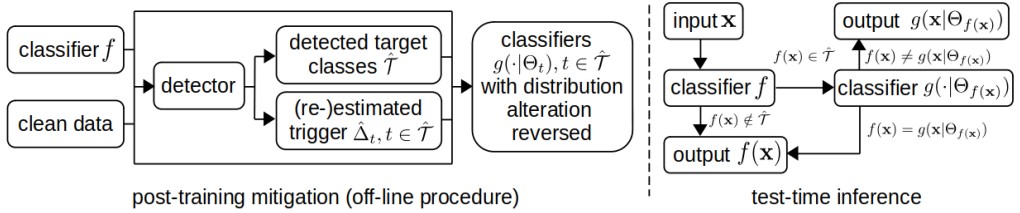

Figure 2: Illustration of our backdoor mitigation framework with a test-time inference rule.

such that $\mathbb{E}[a_j^*(\mathbf{X}_b)] = \mathbb{E}[a_j(\mathbf{X})|Y = -1]$ and $\text{Var}[a_j^*(\mathbf{X}_b)] = \text{Var}[a_j(\mathbf{X})|Y = -1]$ are achieved. But here, we aim to study the quantitative relationship between the distribution divergence and the SIA metric of Def. 3.3 below. Thus, we consider an "intermediate state" with a classifier specified by output node functions $g_-(\cdot|\alpha) : \mathbb{R}^d \to \mathbb{R}$ and $g_+(\cdot|\alpha) : \mathbb{R}^d \to \mathbb{R}$, where for each output node $i \in \{-,+\}$, $g_i(\boldsymbol{x}|\alpha) = \boldsymbol{u}_i^\top \hat{\boldsymbol{a}}(\boldsymbol{x}|\alpha)$ depends on a "transition variable" $\alpha \in [0,1]$, with $\boldsymbol{u}_i$ the weight vector for the original output function $f_i$. $\hat{\boldsymbol{a}}(\boldsymbol{x}|\alpha) = [\hat{a}_1(\boldsymbol{x}|\alpha), \cdots, \hat{a}_J(\boldsymbol{x}|\alpha)]^\top$ is the activation vector for input $\boldsymbol{x}$ where $\hat{a}_j(\boldsymbol{x}|\alpha) = (\boldsymbol{w}_j^\top \boldsymbol{x} - \hat{m}_j(\alpha))\gamma_j/\sqrt{\hat{v}_j(\alpha)} + \beta_j$, with $\hat{m}_j(\alpha) = \alpha m_j + (1-\alpha)m_j^*$ and $\hat{v}_j(\alpha) = (\alpha\sqrt{v_j} + (1-\alpha)\sqrt{v_j^*})^2$ being the "intermediate" mean and variance respectively. Given these settings, our main theoretical results are presented below.

**Definition 3.3. (Source inference accuracy (SIA))** *SIA is the probability that a triggered instance is classified to its original source class (Li et al. (2022a)), i.e. $P[g_-(\mathbf{X}_b|\alpha) > g_+(\mathbf{X}_b|\alpha)]$ here.*

**Theorem 3.1. (Monotonicity of SIA with Divergence)** *If the binary classifier with $f_-$ and $f_+$ is $\eta$-erroneous with $\eta < 1/2$, the attack is $\psi$-successful with $\psi > 1/2$, and $\sigma_b \leq \sigma$, then SIA of the modified classifier, i.e. $P[g_-(\mathbf{X}_b|\alpha) > g_+(\mathbf{X}_b|\alpha)]$, monotonically decreases as $\alpha \in [0,1]$ increases.*

The proof of the theorem is given in Apdx. A.2. Note that the assumptions for Thm. 3.1 are very mild and reasonable. For example, $\eta < 1/2$ is a minimum requirement for the classifier and $\psi > 1/2$ is a minimum requirement for a successful backdoor attack. Moreover, $\sigma_b \leq \sigma$ generally holds empirically since trigger embedding (e.g., consider a patch attack) typically reduces the variance of source class instances (while additive attacks do not change the variance). Also note that $\alpha$ merely gives a way of quantifying distribution divergence for purpose of analysis. According to these results, the core part of our proposed backdoor mitigation approach should be to find a modified classifier $g(\cdot|\boldsymbol{\Theta})$ by minimizing (e.g., using sub-gradient methods) a measure of distribution divergence over a well-chosen *set* of parameters, $\boldsymbol{\Theta}$. This approach is next explicated.

## 4 REVERSING DISTRIBUTION ALTERATION FOR BACKDOOR MITIGATION

### 4.1 PROBLEM DESCRIPTION

**Threat model.** For input space $\mathcal{X}$ and label space $\mathcal{C}$, a classifier that has been successfully backdoor-attacked will predict to the attacker's target class $t^* \in \mathcal{C}$ when a test instance $\boldsymbol{x} \in \mathcal{X}$ is embedded with the backdoor trigger using an incorporation function $\Delta : \mathcal{X} \to \mathcal{X}$. In addition to this "*all-to-one*" setting, we also consider the "*all-to-all*" setting where a test instance from any class $c \in \mathcal{C}$ will be (mis)classified to class $(c + 1)mod|\mathcal{C}|$ when it is embedded with the trigger (Gu et al. (2019)).

**Defender's goal.** Given a trained classifier $f : \mathcal{X} \to \mathcal{C}$ that may possibly be attacked, the defender aims to mitigate possible attacks by producing a mapping $\hat{f} : \mathcal{X} \to \mathcal{C}$ which (a) has high accuracy in classifying clean instances, and (b) when there is a backdoor attack, classifies triggered instances to their original source class, as though there is no trigger embedded, i.e., achieves a high SIA.

**Defender's assumptions.** We consider a *post-training* scenario where the defender has *no access* to the training set of the classifier. The defender does possess an independent clean dataset, but this dataset is *too small* to train an accurate classifier from scratch, and even too small to effectively fine-tune the full set of classifier parameters (Liu et al. (2018a); Zeng et al. (2022); Wang et al. (2019)). The defender has white box access to the classifier, but does not know whether it has been attacked and, if so, does not know the trigger pattern that was used, i.e., the defense is unsupervised.

### 4.2 METHOD

Based on Thm. 3.1, it would seem that a good mitigation approach involves modifying the classifier $f$, i.e. creating a new classifier $g(\cdot|\boldsymbol{\Theta}) : \mathcal{X} \to \mathcal{C}$ from $f$ by applying a transformation function

$h_{j,l}(\cdot|\boldsymbol{\theta}_{j,l}) : \mathbb{R} \to \mathbb{R}$ to the activation of each neuron $j \in \{1, \cdots, J_l\}$ in each layer $l \in \{1, \cdots, L\}$. The parameters $\boldsymbol{\Theta} = \{\boldsymbol{\theta}_{j,l}\}$ should be jointly chosen so as to minimize the aggregation (e.g. sum) of the divergences between the distributions $q_{j,l}(\boldsymbol{\theta}_{<l} \cup \boldsymbol{\theta}_{j,l})$ obtained using $h_{j,l}(\hat{z}_{j,l}(\Delta(\mathbf{X})|\boldsymbol{\theta}_{<l})|\boldsymbol{\theta}_{j,l})$ and the target distributions $p_{j,l}$ for $z_{j,l}(\mathbf{X})$ for $\forall j, l$, where $\mathbf{X}$ follows the clean data distribution, i.e.:

$$\underset{\boldsymbol{\Theta}=\{\boldsymbol{\theta}_{j,l}\}}{\text{minimize}} \quad \sum_{j,l} D_k\big(p_{j,l}||q_{j,l}(\boldsymbol{\theta}_{<l} \cup \boldsymbol{\theta}_{j,l})\big) \tag{3}$$

where: $z_{j,l} : \mathcal{X} \to \mathbb{R}$ and $\hat{z}_{j,l} : \mathcal{X} \to \mathbb{R}$ are activation functions for neuron $j$ in layer $l$ for classifiers $f$ and $g(\cdot|\boldsymbol{\Theta})$ respectively; $\boldsymbol{\theta}_{<l} = \{\boldsymbol{\theta}_{j,l'}|l' < l\}$ represents all transformation parameters prior to layer $l$; $D_k(p||q) := \mathbb{E}_q[k(p/q)]$ for a convex function $k : [0, \infty) \to \mathbb{R}$ satisfying $k(1) = 0$ and belonging to the family of $f$-divergences for any distributions $p$ and $q$ (Ali & Silvey (1966)).

However, in practice, we will face the following challenges. **Challenge 1:** The defender does not know *a priori* whether there is an attack. When there is no attack, no distribution correction should be needed. Moreover, when there is an attack, while the classifier $g(\cdot|\boldsymbol{\Theta})$ with optimal transformation functions for neuron activation will achieve a high SIA on triggered instances, its accuracy on *clean* instances (especially those not from the backdoor target class) may be degraded[2]. **Challenge 2:** If there is an attack, the attack setting, i.e. all-to-one or all-to-all, and the ground-truth backdoor trigger $\Delta$ are both unknown to the defender. **Challenge 3:** The density form for $z_{j,l}(\Delta(\mathbf{X}))$ may get altered by the trigger $\Delta$ and will likely be different from the density form for $z_{j,l}(\mathbf{X})$ – moreover both will likely be non-Gaussian. Thus, (3) cannot be easily minimized, e.g., simply by matching the mean and variance.

For *Challenges 1&2*, we leverage existing *post-training* backdoor detection approaches to infer: whether the classifier $f$ is backdoor attacked and the associated target classes when $f$ is attacked (Wang et al. (2019); Chen et al. (2019b); Liu et al. (2019)). These detectors, following the same assumptions in Sec. 4.1, reverse-engineer a trigger for each *putative target class* on the clean dataset possessed by the defender. Then anomaly detection is performed on statistics derived from these reverse-engineered triggers, e.g. the estimated size for patch triggers used by Wang et al. (2019).

Here, the detector in our framework is different from most existing ones in order to cover a broad range of attack settings including *all-to-one* and *all-to-all* attacks. We first reverse-engineer a trigger by solving an optimization problem defined on the clean set to get a detection statistic for each ordered *putative class pair* $(s,t) \in \mathcal{C} \times \mathcal{C}$. For the Xiang et al. (2020) method, this statistic is (the reciprocal of) the estimated perturbation size inducing high (mis)classifications from $s$ to $t$. For Wang et al. (2019), it is the estimated patch size inducing high (mis)classifications from $s$ to $t$. Then we apply the anomaly detection approach in Wang et al. (2019), based on the MAD criterion (Hampel (1974)), to all the obtained statistics to find *all* the *outlier* statistics. We denote the set of detected class pairs associated with these outlier statistics as $\hat{\mathcal{P}}$, and denote $\hat{\mathcal{T}} = \{t \in \mathcal{C} \mid \exists s \in \mathcal{C} \text{ s.t. } (s,t) \in \hat{\mathcal{P}}\}$ as the set of detected target classes. For *each* $t \in \hat{\mathcal{T}}$, we (re-)estimate a trigger $\hat{\Delta}_t$ (as a *surrogate* for the true backdoor trigger, which is unknown) using clean instances from all detected source classes[3] $\hat{\mathcal{S}}(t) = \{s \in \mathcal{C}|(s,t) \in \hat{\mathcal{P}}\}$. Then, *for each detected target class* $t \in \hat{\mathcal{T}}$, we construct a classifier $g(\cdot|\boldsymbol{\Theta}_t)$ by solving the distribution divergence minimization problem using its (re-)estimated $\hat{\Delta}_t$.

For any test input $\boldsymbol{x} \in \mathcal{X}$, if classifier $f$ is deemed attack-free, i.e. $\hat{\mathcal{P}} = \emptyset$, the classification output under our mitigation framework will be $\hat{f}(\boldsymbol{x}) = f(\boldsymbol{x})$. Otherwise, if $f(\boldsymbol{x}) \in \mathcal{C} \setminus \hat{\mathcal{T}}$, we trust the class decision and set $\hat{f}(\boldsymbol{x}) = f(\boldsymbol{x})$ both because $\boldsymbol{x}$ is unlikely to possess a trigger and because a successful attack should not degrade the classifier's accuracy on clean instances. However, if $f(\boldsymbol{x}) = t \in \hat{\mathcal{T}}$, there are two main possibilities: 1) $\boldsymbol{x}$ is a clean instance truly from class $t$; 2) $\boldsymbol{x}$ is classified to class $t$ due to the presence of the trigger. To distinguish these two cases, we feed $\boldsymbol{x}$ to the optimized $g(\cdot|\boldsymbol{\Theta}_t)$. If $g(\boldsymbol{x}|\boldsymbol{\Theta}_t) \neq f(\boldsymbol{x})$, $\boldsymbol{x}$ likely contains a trigger, and thus we should set $\hat{f}(\boldsymbol{x}) = g(\boldsymbol{x}|\boldsymbol{\Theta}_t)$, which is likely the original source class of $\boldsymbol{x}$ based on our theoretical results. The outline of our mitigation framework is summarized in Fig. 2. Note that in the test-time inference procedure above, the major (additional) computation for both backdoor trigger instance detection and source class inference is a forward propagation for feeding $\boldsymbol{x}$ to $g(\cdot|\boldsymbol{\Theta}_t)$, which is comparable

---

[2]For example, the histogram for clean activations in Fig. 1b will be shifted away (to the left).

[3]More reliable trigger estimation can be achieved in this way for a detected target class.

to the computation required for classification using $f$. Moreover, such additional computation occurs only if an attack is detected and $f(\boldsymbol{x}) = t$; thus, our test-time inference is very efficient.

Now the remaining problem is to address *Challenge 3*, which is critical to the estimation of $\boldsymbol{\Theta}_t$ using the reverse-engineered trigger $\hat{\Delta}_t$ *for each detected target class* $t \in \hat{\mathcal{T}}$. For simplicity, we will drop the subscript $t$ below without loss of generality. Our main goals are: (a) specifying the structure of the transformation function $h_{j,l}$ with its associated parameters $\boldsymbol{\theta}_{j,l}$, (b) empirical estimation of the distribution divergence in Eq. (3) using a clean dataset (i.e. the subset of clean instances from classes in $\hat{\mathcal{S}}(t)$ for each detected class $t$), and (c) choosing the convex function $k$ for the divergence form. For (a), we consider the following transformation function with parameters $\boldsymbol{\theta}_{j,l} = \{\mu_{j,l}, \sigma_{j,l}, \upsilon_{j,l}, \omega_{j,l}\}$:

$$h_{j,l}(z) = \max\{\min\{\frac{z - \mu_{j,l}}{\sigma_{j,l}}, \omega_{j,l}\}, \upsilon_{j,l}\} \tag{4}$$

where $\mu_{j,l}$ and $\sigma_{j,l}$ specify the location and scale of the activation distribution, respectively, while $\upsilon_{j,l}, \omega_{j,l}$ control the shape of the tail of the distribution. For goal (b), we quantize the real line into $M$ intervals $\mathcal{I}_1 = (-\infty, b_1), \mathcal{I}_2 = [b_1, b_2), \cdots, \mathcal{I}_M = [b_{M-1}, \infty)$, for $M$ sufficiently large. Then the distribution divergence in Eq. (3) for each node $j$ and layer $l$ is computed on discrete distributions $\hat{p}_{j,l}$ and $\hat{q}_{j,l}$ over these intervals. Specifically, the discrete distributions are estimated using a subset $\mathcal{D}_t$ of instances from classes $\hat{\mathcal{S}}(t)$, with the probabilities for interval $\mathcal{I}_i$ computed by:

$$\hat{p}_{j,l}^{(i)} = \frac{1}{|\mathcal{D}_t|} \sum_{\boldsymbol{x} \in \mathcal{D}_t} \mathbb{1}[z_{j,l}(\boldsymbol{x}) \in \mathcal{I}_i] \quad \text{and} \quad \hat{q}_{j,l}^{(i)} = \frac{1}{|\mathcal{D}_t|} \sum_{\boldsymbol{x} \in \mathcal{D}_t} \mathbb{1}[h_{j,l}(\hat{z}_{j,l}(\hat{\Delta}_t(\mathbf{X})|\boldsymbol{\theta}_{<l})|\boldsymbol{\theta}_{j,l}) \in \mathcal{I}_i]. \tag{5}$$

To ensure that the distribution divergence is differentiable with reference to the parameters, such that it can be minimized using (e.g.) gradient descent, we approximate the non-differentiable indicator function $\mathbb{1}[\cdot]$ in Eq. (5) using differentiable functions such as the sigmoid, i.e. we redefine:

$$\mathbb{1}[z \in \mathcal{I}_i] = sigmoid(\tau(z - b_{i-1})) - sigmoid(\tau(z - b_i)) \tag{6}$$

where $\tau$ is a scale factor controlling the error of approximation. For $\mathcal{I}_1$ and $\mathcal{I}_M$, which have semi-infinite support, we use a single sigmoid in Eq. (6). The choice of the intervals and $\tau$ is not critical to the performance, as long as the length of the finite intervals is sufficiently small, as will be shown in Tab. 4 in Sec. 5. Finally, for goal (c), we consider several different divergence forms including the total variation (TV) divergence with $k(r) = |r - 1|/2$, the Jensen-Shannon (JS) divergence with $k(r) = r \log \frac{2r}{r+1} + \log \frac{2}{r+1}$, and the Kullback-Leibler (KL) divergence with $k(r) = r \log r$. The choice of the divergence form is also not critical to the mitigation performance (see Apdx. E).

## 5 EXPERIMENTS

### 5.1 EXPERIMENT SETUP

**Datasets**: Our main experiments are conducted on the benchmark CIFAR-10 dataset, which contains 60,000 $32 \times 32$ color images from 10 classes, with 5,000 images per class for training and 1,000 images per class for testing (Krizhevsky & Hinton (2009)). We also show the effectiveness of our proposed mitigation framework on other benchmark datasets including GTSRB (Houben et al. (2013)), CIFAR-100 (Krizhevsky & Hinton (2009)), ImageNette (Howard (2020)), and TinyImageNet. Details of these datasets can be found in Apdx. B.1. Data allocation in our experiments strictly follows the assumptions in Sec. 4.1. For each dataset, we randomly sample 10% of the test set to form the small, clean dataset $\mathcal{D}_{\text{Defense}}$ assumed for the defender. The remaining test instances, denoted by $\mathcal{D}_{\text{Test}}$, are reserved for performance evaluation.

**Attack settings**: In this paper, we consider standard backdoor attacks launched by poisoning the training set of the classifier (Gu et al. (2019); Chen et al. (2017)). In particular, we consider both the *all-to-one* (**A2O**) attacks and the *all-to-all* (**A2A**) attacks in our main experiments on CIFAR-10. For A2O attacks on CIFAR-10, we arbitrarily choose class 9 as the target class; while for A2A attacks, as described in Sec. 4.1, triggered instances from any class $c \in \mathcal{C}$ are supposed to be (mis)classified to class $(c + 1) mod |\mathcal{C}|$. For each attack setting, we consider the following triggers: 1) a $3 \times 3$ random patch (**BadNet**) with a randomly selected location (fixed for all triggered images for each attack) used in Gu et al. (2019); 2) an additive perturbation (with size 2/255) resembling a chessboard (**CB**) used in Xiang et al. (2020); 3) a single pixel (**SP**) perturbed by 75/255 with a randomly selected location (fixed for all triggered images for each attack) used by Tran et al.

| Trigger type | | BadNet | | CB | | $l_0$ inv | | $l_2$ inv | | SP | | WaNet | |
|---|---|---|---|---|---|---|---|---|---|---|---|---|---|
| | | A2O | A2A | A2O | A2A | A2O | A2A | A2O | A2A | A2O | A2A | A2O | A2A |
| Vanilla | ACC | 0.9122 | 0.9121 | 0.9135 | 0.9098 | 0.9135 | 0.9131 | 0.9130 | 0.9126 | 0.9138 | 0.9060 | 0.9032 | 0.8994 |
| | ASR | 0.9573 | 0.8658 | 0.9685 | 0.8692 | 0.9989 | 0.8973 | 0.9889 | 0.8620 | 0.8912 | 0.8550 | 0.9153 | 0.8216 |
| | SIA | 0.0397 | 0.0432 | 0.0293 | 0.0257 | 0.0010 | 0.0151 | 0.0107 | 0.0194 | 0.1016 | 0.0772 | 0.0772 | 0.0714 |
| NC | ACC | 0.8797 | 0.8762 | 0.8735 | 0.8776 | 0.8835 | 0.8767 | 0.8750 | 0.8690 | 0.8854 | 0.8614 | 0.8748 | 0.8756 |
| | ASR | 0.0130 | **0.0154** | **0.0064** | 0.0155 | 0.0120 | 0.0150 | 0.0080 | 0.0179 | 0.0335 | 0.0188 | 0.0144 | 0.1381 |
| | SIA | 0.8532 | 0.8614 | 0.8312 | 0.8597 | 0.8654 | 0.8650 | 0.7932 | 0.8254 | 0.8362 | 0.8477 | 0.8231 | 0.7183 |
| I-BAU | ACC | 0.8500 | 0.8758 | 0.8812 | 0.8719 | 0.8452 | 0.8800 | 0.8825 | 0.8726 | 0.8666 | 0.8745 | 0.8777 | 0.8700 |
| | ASR | **0.0094** | 0.0164 | 0.1973 | 0.0811 | 0.0091 | 0.0133 | 0.2600 | 0.3353 | 0.0172 | **0.0154** | 0.1339 | 0.1253 |
| | SIA | 0.8301 | 0.8583 | 0.6399 | 0.7756 | 0.8277 | 0.8673 | 0.5549 | 0.4928 | 0.8479 | 0.8609 | 0.7059 | 0.7269 |
| ANP | ACC | 0.8644 | 0.8492 | 0.8241 | 0.8577 | 0.8455 | 0.8648 | 0.8345 | 0.8421 | 0.8195 | 0.8411 | 0.8298 | 0.8607 |
| | ASR | 0.0474 | 0.1199 | 0.3351 | 0.0927 | 0.0836 | 0.1326 | 0.4703 | 0.2648 | 0.1229 | 0.0495 | 0.0263 | 0.0835 |
| | SIA | 0.8184 | 0.7205 | 0.4587 | 0.7168 | 0.7697 | 0.7324 | 0.3351 | 0.4976 | 0.7060 | 0.7942 | 0.7368 | 0.7451 |
| NAD | ACC | 0.8814 | 0.8819 | 0.8800 | **0.8908** | 0.8958 | **0.9047** | 0.8991 | **0.8781** | 0.8813 | 0.8761 | 0.8592 | **0.8963** |
| | ASR | 0.0193 | 0.7132 | 0.0871 | 0.0681 | 0.0356 | 0.0457 | 0.0254 | 0.0191 | 0.0667 | 0.0647 | 0.0571 | 0.1056 |
| | SIA | 0.8498 | 0.1520 | 0.7711 | 0.8084 | 0.8504 | 0.8534 | 0.8221 | 0.8337 | 0.8123 | 0.8082 | 0.7710 | 0.7773 |
| ARGD | ACC | 0.8689 | 0.8482 | 0.8800 | 0.8774 | 0.8880 | 0.8885 | 0.8669 | 0.8583 | 0.8899 | 0.8728 | 0.8739 | 0.8755 |
| | ASR | 0.0368 | 0.0839 | 0.0099 | **0.0117** | 0.0079 | 0.0122 | 0.0125 | 0.0179 | 0.0955 | 0.0452 | 0.0111 | 0.0362 |
| | SIA | 0.8217 | 0.7544 | 0.8657 | 0.8690 | 0.8725 | 0.8786 | 0.8168 | 0.8297 | 0.7934 | 0.8295 | 0.8283 | 0.8241 |
| BNA (ours) | ACC | **0.9032** | **0.8951** | **0.9072** | 0.8615 | **0.9068** | 0.8944 | **0.9005** | 0.8638 | **0.9058** | 0.8921 | **0.8945** | 0.8792 |
| | ASR | 0.0139 | 0.0189 | 0.0127 | 0.0202 | **0.0033** | **0.0111** | **0.0042** | 0.0168 | 0.0104 | 0.0225 | **0.0041** | 0.0191 |
| | SIA | **0.8835** | **0.8841** | **0.8787** | 0.8820 | **0.8924** | **0.8942** | 0.8383 | 0.8522 | **0.8863** | **0.8811** | 0.8530 | 0.8607 |

Table 1: Average ACC, ASR, and SIA for our BNA, compared with NC, NAD, I-BAU, ANP, and ARGD, against all the created attacks applied to ResNet-18 trained on the CIFAR-10 dataset.

(2018); 4) invisible triggers generated with $l_0$ and $l_2$ norm constraints ($l_0$ **inv** and $l_2$ **inv** respectively) proposed by Li et al. (2021a); 5) a warping-based trigger (**WaNet**) proposed by Nguyen & Tran (2021). Details for generating these triggers are deferred to Apdx. B.3. For experiments on CIFAR-10, we randomly create 5 attacks for each attack setting and each trigger (e.g. with random location). For experiments on the other four datasets, we only consider A2O attacks for a subset of triggers where sufficiently high success rate can be achieved. For each dataset, we create one attack for each trigger being considered. A2A attacks are not considered for these datasets since they are not successful due to the insufficiency of data. More details about the attacks, including the number of backdoor-trigger images used for poisoning and the target class selected to create A2O attacks for the four datasets other than CIFAR-10, are shown in Apdx. B.3.

**Performance evaluation metrics**: 1) The attack success rate (**ASR**) is the fraction of clean instances in $\mathcal{D}_{\text{Test}}$ (mis)classified to the designated target class when the backdoor trigger is embedded. 2) The clean test accuracy (**ACC**) is the DNN's accuracy on $\mathcal{D}_{\text{Test}}$ without trigger embedding. 3) The **SIA** (Def 3.3) is the fraction of clean instances in $\mathcal{D}_{\text{Test}}$ classified to the original source class when the trigger is embedded. For a successful backdoor attack, ASR and ACC should be large, while SIA should be small. For a successful mitigation approach, the resulting ASR should be small, while ACC and SIA should be large.

**Training settings**: We train one classifier for each attack to evaluate our mitigation approach against existing ones. Training configurations, including the DNN architecture, batch size, number of epochs, etc., are detailed in Tab. 6 in Apdx. B.2. Data augmentation choices, including random cropping and horizontal flipping, are applied to each training instance. As shown in Tab. 1, the defenseless "vanilla" classifiers being attacked achieve high ACC but suffer high ASR and low SIA (averaged over the five attacks we created) for all trigger types and for both A2O and A2A settings, i.e., the attacks are all successful and hence adequate for performance evaluation.

## 5.2 BACKDOOR MITIGATION RESULTS

We compare our mitigation approach (named **BNA** in the sequel) with five well-known and/or state-of-the-art methods, including NC(Wang et al. (2019)), NAD(Li et al. (2021c)), I-BAU(Zeng et al. (2022)), ANP(Wu & Wang (2021)), and ARGD(Xia et al. (2022)). For all these other methods, we used their officially posted code for implementation. For our BNA, following Sec. 4.2, we first perform detection by reverse-engineering a backdoor trigger for each class pair using objective functions from, *e.g.*, Wang et al. (2019); Xiang et al. (2020) and then feeding the statistics obtained from the estimated trigger to an anomaly detector. Our anomaly detector is based on MAD, which is a classical approach also used by Wang et al. (2019); Chen et al. (2019b); Wang et al. (2020). Here, we set the detection threshold at "7-MAD" which easily catches all the backdoor class pairs. More details, including pattern estimation and detection statistics are shown in Apdx. C due to space limitation. Then, for each detected target class, we solve the divergence minimization problem

| | the number of poisoned instances per class | | | | | perturbation size (*255) | | | | |
|---|---|---|---|---|---|---|---|---|---|---|
| | 50 | 100 | 150 | 200 | 250 | 2 | 3 | 4 | 5 | 6 |
| ACC | 0.9112 | 0.9094 | 0.9098 | 0.9102 | 0.9015 | 0.9094 | 0.9041 | 0.9079 | 0.8992 | 0.8912 |
| ASR | 0.0095 | 0.0141 | 0.0121 | 0.0170 | 0.0090 | 0.0141 | 0.0395 | 0.0222 | 0.0109 | 0.0388 |
| SIA | 0.8851 | 0.8837 | 0.8728 | 0.8840 | 0.8662 | 0.8851 | 0.8783 | 0.8711 | 0.8814 | 0.8435 |

Table 2: ACC, ASR, and SIA for our BNA as a function of (1) the number of poisoned instances injected into the training set; (2) the perturbation size under all-to-one CB attack.

to optimize the transformation functions using learning rate 0.01 for 30 epochs. If a neuron is followed by a BN (which is very common), instead of applying an additional transformation function $h_{j,l}$, we treat the mean and standard deviation of BN as the parameters $\mu_{j,l}$ and $\sigma_{j,l}$ associated with $h_{j,l}$ respectively. Here, we only show results for BNA with the total variation divergence. Results for KL-divergence and JS-divergence are deferred to Apdx. E. To compute the divergence, we use the "interval trick" (Eq. (5)) to obtain the discrete empirical distribution. For simplicity, we let all finite intervals, $\mathcal{I}_i = [b_{i-1}, b_i), i = 1, \cdots, M$, have the same length $\Delta b = 0.1$. For each neuron, we set $b_{\min}$ and $b_{\max}$ as the minimum and maximum activations, respectively, when feeding in clean instances from $\mathcal{D}_{\text{Defense}}$ to the poisoned classifier $f$. Then, the number of intervals is $M = \lceil \frac{b_{\max} - b_{\min}}{\Delta b} \rceil$; and all intervals can be specified by $b_0 = b_{\min}$ and $b_i = b_{i-1} + \Delta b$. Finally, the scale factor in Eq. (6) is set to $\tau = 150$, which is obtained by line search to minimize the total variation between the "soft" distribution and the empirical one. In fact, the choices for $\Delta b$ and $\tau$ (over reasonable ranges) has little impact on our mitigation performance, as shown in Tab. 4.

In Tab. 1, we show the ASR, ACC, and SIA for our BNA compared with the other five methods (which are all tuning-based) for attacks on CIFAR-10. Each metric is averaged over the five attacks created for each trigger type and attack setting, with the highest ACC and SIA, and the lowest ASR in bold. Although the five tuning-based methods we compare here can effectively deactivate the backdoor attacks (*i.e.*, significantly reduce ASRs), there is a clear drop (3%-15%) in both ACC and SIA. This is possibly due to tuning many DNN parameters using very limited data. Moreover, we found these tuning-based methods are sensitive to the choices of hyper-parameters, such as the learning rate. For ANP with neuron pruning, the performance is acceptable only for A2O with the BadNet trigger. One possible reason is that invisible, perturbation-based triggers affect most neurons only moderately (Wang et al. (2022a)); thus, pruning a small number of neurons cannot mitigate the attack. In contrast, our method successfully mitigates all these backdoor attacks (with generally the best ACC and ASR compared with the others) regardless of the trigger type and attack setting. Notably, *since the purpose of BNA's divergence minimization is to maximize the SIA*, it unsurprisingly achieves the best SIA with a clear margin over all other methods, in all cases (the corresponding distribution divergnces are shown in Tab. 9). We also tune the poisoning ratio and perturbation size used in A2O CB attacks, and the performance for our BNA slightly declines as the attack is strengthened, as shown in Tab. 2. However, it still outperforms the other methods (see Tab. 11 in Apdx. F).

*Why tuning-based methods like NC cannot achieve SIA as high as our BNA without parameter altering?* Note that NC tunes the classifier using instances embedded with the estimated trigger but without label flipping. This is equivalent to minimizing the divergence between internal activation distributions for clean and triggered instances, but with the parameters changed. Even for an optimal (zero) divergence, the best achievable SIA of NC is still upper-bounded by the ACC of the classifier after tuning, which usually *drops* due to the data insufficiency. By contrast, the reference distribution for our divergence minimization is obtained by feeding clean instances to the poisoned classifier without changing its parameters; thus, it is a "better" reference with a higher upper-bound ACC.

Results of our BNA on other datasets are shown in Table 3. The ACC for DNNs trained without attack for GTSRB, CIFAR-100, ImageNette, and TinyImageNet are 0.9567, 0.6926, 0.8726, and 0.5224, respectively; while ACC, ASR, and SIA for attacked DNNs are shown in the row "Vanilla". We apply our BNA on the poisoned DNNs, with the same settings as for CIFAR-10, which significantly reduces ASR (to less than 1.3% in all cases), with uniformly high SIA and ACC.

## 5.3 TEST-TIME BACKDOOR-TRIGGER INSTANCES DETECTION

Different from other tuning-based backdoor mitigation approaches, our BNA can also detect backdoor-trigger instances at test-time, as described in Sec. 4.2 and shown in Fig. 2. Here, we evaluate the accuracy of our test-time detection compared with a state-of-the-art detector named

| Trigger type | | GTSRB | | | | | CIFAR-100 | | | | TinyImageNet | ImageNette |
|---|---|---|---|---|---|---|---|---|---|---|---|---|
| | | BadNet | CB | $l_0$ inv | $l_2$ inv | WaNet | BadNet | CB | $l_0$ inv | $l_2$ inv | BadNet | BadNet |
| Vanilla | ACC | 0.9517 | 0.9556 | 0.9531 | 0.9521 | 0.9408 | 0.6796 | 0.6917 | 0.6863 | 0.6804 | 0.5192 | 0.8626 |
| | ASR | 1.0000 | 1.0000 | 1.0000 | 0.9794 | 0.9000 | 0.9037 | 0.9169 | 0.9935 | 0.9097 | 0.8058 | 0.9144 |
| | SIA | 0.0000 | 0.0000 | 0.0000 | 0.0169 | 0.0905 | 0.0781 | 0.0646 | 0.0063 | 0.0707 | 0.1134 | 0.0771 |
| BNA | ACC | 0.9491 | 0.9548 | 0.9505 | 0.9500 | 0.9404 | 0.6770 | 0.6863 | 0.6858 | 0.6787 | 0.5178 | 0.7941 |
| | ASR | 0.0000 | 0.0000 | 0.0122 | 0.0001 | 0.0041 | 0.0002 | 0.0524 | 0.0062 | 0.0016 | 0.0043 | 0.0016 |
| | SIA | 0.9312 | 0.9454 | 0.9330 | 0.8945 | 0.9338 | 0.6526 | 0.5880 | 0.6169 | 0.5224 | 0.4965 | 0.7940 |

Table 3: ACC, ASR, and SIA for our BNA against all-to-one attacks on CIFAR-100, GTSRB, ImageNette, and TinyImageNet datasets.

| $\tau$ ($\Delta b$=0.1) | 10 | 100 | 200 | 300 | 400 | 500 | 600 | 700 | 800 | 900 | 1000 |
|---|---|---|---|---|---|---|---|---|---|---|---|
| ACC | 0.9024 | 0.9025 | 0.9022 | 0.9019 | 0.9024 | 0.9019 | 0.9022 | 0.9018 | 0.9017 | 0.9015 | 0.9021 |
| ASR | 0.0257 | 0.022 | 0.0206 | 0.0207 | 0.0214 | 0.0202 | 0.0212 | 0.0209 | 0.0207 | 0.0201 | 0.0204 |
| SIA | 0.8744 | 0.8758 | 0.8774 | 0.8768 | 0.8768 | 0.8775 | 0.8773 | 0.8768 | 0.8772 | 0.8778 | 0.8777 |
| $\Delta b$ ($\tau$=150) | 0.1 | 0.11 | 0.12 | 0.13 | 0.14 | 0.15 | 0.16 | 0.17 | 0.18 | 0.19 | 0.2 |
| ACC | 0.9028 | 0.9018 | 0.9019 | 0.9023 | 0.9019 | 0.9023 | 0.9019 | 0.9019 | 0.9022 | 0.9022 | 0.9022 |
| ASR | 0.0197 | 0.0202 | 0.0199 | 0.0204 | 0.0199 | 0.0204 | 0.0198 | 0.0206 | 0.0206 | 0.0207 | 0.0212 |
| SIA | 0.8799 | 0.8775 | 0.8779 | 0.8773 | 0.8772 | 0.8779 | 0.8773 | 0.8767 | 0.8779 | 0.8769 | 0.8764 |

Table 4: ACC, ASR, and SIA for our BNA as a function of scale factor and bin size on ResNet-18 trained on CIFAR-10 poisoned by all-to-one BadNet attack.

| Trigger type | | BadNet | | CB | | $l_0$ inv | | $l_2$ inv | | SP | | WaNet | |
|---|---|---|---|---|---|---|---|---|---|---|---|---|---|
| | | A2O | A2A | A2O | A2A | A2O | A2A | A2O | A2A | A2O | A2A | A2O | A2A |
| BNA | FPR | **0.1390** | **0.0606** | **0.1144** | **0.1092** | **0.1413** | **0.0600** | 0.1976 | **0.1027** | **0.1323** | **0.0656** | **0.1406** | **0.0865** |
| | TPR | **0.9872** | **0.9508** | **0.9872** | **0.9682** | 0.9967 | **0.9873** | **0.9958** | **0.9793** | **0.9894** | **0.9294** | **0.9959** | **0.9248** |
| STRIP | FPR | 0.15 | 0.15 | 0.15 | 0.15 | 0.15 | 0.15 | **0.15** | 0.15 | 0.15 | 0.15 | 0.15 | 0.15 |
| | TPR | 0.9638 | 0.5147 | 0.6802 | 0.2089 | **0.9995** | 0.1053 | 0.9924 | 0.5272 | 0.8522 | 0.3123 | 0.0202 | 0.0411 |

Table 5: TPR and FPR for our BNA, compared with STRIP, against all attacks created on CIFAR-10.

STRIP (Gao et al. (2019)). For any input image during inference, STRIP blends it with clean images possessed by the defender. The blended image is fed into the poisoned DNN, with an entropy calculated on the output posteriors. If the entropy is lower than a prescribed detection threshold, the input is deemed to be embedded with the trigger. Here, we set the detection threshold at 15% FPR for STRIP which achieves a generally good trade-off between TPR and FPR. In contrast, our BNA does not need to set a detection threshold. In Tab. 5, we show the True Positive Rate (TPR, *i.e.*, the fraction of backdoor-trigger images correctly detected) and the False Positive Rate (FPR, *i.e.*, the fraction of clean test images from the backdoor target class(es) that are falsely detected) for both methods. Although STRIP performs well on A2O attacks for some trigger types, *e.g.*, BadNet, $l_0$ inv, and $l_2$ inv, its TPR drops drastically on attacks using human-imperceptible triggers, especially the WaNet attacks. Moreover, it does not perform well on all A2A attacks, with largest TPR of only 0.5272. By contrast, our BNA is effective for all these attacks – it detects almost all the backdoor-trigger images, with FPRs comparable to STRIP.

## 5.4 MITIGATION PERFORMANCE AGAINST ADAPTIVE ATTACKS

A recent backdoor attack proposed by Doan et al. (2021) minimizes a metric similar to our activation distribution alteration, in order to achieve better stealthiness. This attack can be viewed as an adaptive attack against our mitigation defense since the trained classifier will be more sensitive to even a smaller distribution divergence than for ordinary backdoor attacks. Nevertheless, our method successfully mitigates this attack. In our experiment on CIFAR-10, the average distribution total variation divergence over all neurons is reduced from 8067 to 2789. Accordingly, the ACC/ASR before and after mitigation are 0.9162/0.9978 and 0.8906/0.0072 respectively, with SIA 0.8496.

## 6 CONCLUSION

In this paper, we revealed an activation distribution alteration property for backdoor attacks. We found that by correcting such alteration, backdoor trigger instances will be classified to their original source classes. Accordingly, we proposed a backdoor mitigation approach without changing any parameters of the classifier, which outperformed methods that use DNN fine-tuning. Moreover, our method can detect instances with the trigger during inference.

## ETHICS STATEMENT

The main purpose of this research is to understand the behavior of deep learning systems facing malicious activities, and enhance their safety level. The backdoor attack considered in this paper is well-known, with open-sourced implementation code. Thus, publication of this paper will be beneficial to the community in defending against backdoor attacks. The code of our defense will be released if the paper is accepted.

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

# A    PROOF OF THEOREMS IN THE MAIN PAPER

## A.1    DERIVATION OF EQ. (2)

Here, we provide the derivation showing that $m_j^*$ and $v_j^*$ in Eq. (2) are the solutions to:

$$\mathbb{E}[a_j^*(\mathbf{X}_b)] = \mathbb{E}[a_j(\mathbf{X})|Y = -1] \tag{7}$$

$$\mathrm{Var}[a_j^*(\mathbf{X}_b)] = \mathrm{Var}[a_j(\mathbf{X})|Y = -1] \tag{8}$$

Based on Eq. (1), the above equations can be expanded as the following:

$$\mathbb{E}[\frac{\boldsymbol{w}_j^\top \mathbf{X}_b - m_j^*}{\sqrt{v_j^*}}\gamma_j + \beta_j] = \mathbb{E}[\frac{\boldsymbol{w}_j^\top \mathbf{X} - m_j}{\sqrt{v_j}}\gamma_j + \beta_j | Y = -1] \tag{9}$$

$$\mathrm{Var}[\frac{\boldsymbol{w}_j^\top \mathbf{X}_b - m_j^*}{\sqrt{v_j^*}}\gamma_j + \beta_j] = \mathrm{Var}[\frac{\boldsymbol{w}_j^\top \mathbf{X} - m_j}{\sqrt{v_j}}\gamma_j + \beta_j | Y = -1] \tag{10}$$

We first solve Eq. (10) for $(\mathbf{X}|Y = -1) \sim \mathcal{N}(-\boldsymbol{\mu}, \sigma^2 \boldsymbol{I})$ and $\mathbf{X}_b \sim \mathcal{N}(\boldsymbol{\mu}_b, \sigma_b^2 \boldsymbol{I})$, which leads to:

$$v_j^* = \frac{\sigma_b}{\sigma}v_j \tag{11}$$

By substituting Eq. (11) into Eq. (9), and since $\boldsymbol{\mu}_b = -\boldsymbol{\mu} + \boldsymbol{\epsilon}$, we get the following:

$$m_j^* = \sqrt{\frac{v_j^*}{v_j}}(m_j - \boldsymbol{w}_j^\top \boldsymbol{\mu}) + \boldsymbol{w}_j^\top \boldsymbol{\mu}_b$$

$$= \frac{\sigma_b}{\sigma}m_j + (\frac{\sigma_b}{\sigma} - 1)\boldsymbol{w}_j^\top \boldsymbol{\mu} + \boldsymbol{w}_j^\top \boldsymbol{\epsilon}.$$

## A.2    PROOF OF THEOREM 3.1

*Proof.* First, let's specify the following vector/matrix representations that will be used throughout this proof:

$$\boldsymbol{W} = [\boldsymbol{w}_1, \cdots, \boldsymbol{w}_J]^\top \in \mathbb{R}^{J \times d}$$

$$\boldsymbol{V} = \begin{bmatrix} v_1 & \cdots & 0 \\ \vdots & \ddots & \vdots \\ 0 & \cdots & v_J \end{bmatrix} \in \mathbb{R}^{J \times J} \qquad \hat{\boldsymbol{V}}(\alpha) = \begin{bmatrix} \hat{v}_1(\alpha) & \cdots & 0 \\ \vdots & \ddots & \vdots \\ 0 & \cdots & \hat{v}_J(\alpha) \end{bmatrix} \in \mathbb{R}^{J \times J}$$

$$\boldsymbol{m} = \begin{bmatrix} m_1 \\ \vdots \\ m_J \end{bmatrix} \in \mathbb{R}^J \quad \hat{\boldsymbol{m}}(\alpha) = \begin{bmatrix} \hat{m}_1(\alpha) \\ \vdots \\ \hat{m}_J(\alpha) \end{bmatrix} \in \mathbb{R}^J \quad \boldsymbol{\Gamma} = \begin{bmatrix} \gamma_1 & \cdots & 0 \\ \vdots & \ddots & \vdots \\ 0 & \cdots & \gamma_J \end{bmatrix} \in \mathbb{R}^{J \times J} \quad \boldsymbol{\beta} = \begin{bmatrix} \beta_1 \\ \vdots \\ \beta_J \end{bmatrix} \in \mathbb{R}^J$$

$$\boldsymbol{a}(\cdot) = \begin{bmatrix} a_1(\cdot) \\ \vdots \\ a_J(\cdot) \end{bmatrix} \in \mathbb{R}^J \quad \hat{\boldsymbol{a}}(\cdot|\alpha) = \begin{bmatrix} \hat{a}_1(\cdot|\alpha) \\ \vdots \\ \hat{a}_J(\cdot|\alpha) \end{bmatrix} \in \mathbb{R}^J \quad \boldsymbol{a}^*(\cdot) = \begin{bmatrix} a_1^*(\cdot) \\ \vdots \\ a_J^*(\cdot) \end{bmatrix} \in \mathbb{R}^J$$

Let $\mathbf{X}_- = (\mathbf{X}|Y = -1) \sim \mathcal{N}(-\boldsymbol{\mu}, \sigma^2 \boldsymbol{I})$ denote a random instances from the source class '−1' for simplicity. Let $\boldsymbol{u} = \boldsymbol{u}_- - \boldsymbol{u}_+$ with $\boldsymbol{u}_-$ and $\boldsymbol{u}_+$ being the weight vectors associated with the node for class '−1' and the node for class '+1' respectively. Then, it is easy to see that:

$$\hat{\boldsymbol{a}}(\mathbf{X}_b|\alpha)\big|_{\alpha=1} = \boldsymbol{a}(\mathbf{X}_b) \quad \text{and} \quad \hat{\boldsymbol{a}}(\mathbf{X}_b|\alpha)\big|_{\alpha=0} = \boldsymbol{a}^*(\mathbf{X}_b),$$

and taking one step further by setting $\alpha = 1$, we have the following:

$$P[g_-(\mathbf{X}_b|\alpha) > g_+(\mathbf{X}_b|\alpha)\big|\alpha = 1] = P[\boldsymbol{u}^\top \hat{\boldsymbol{a}}(\mathbf{X}_b|\alpha) > 0\big|\alpha = 1] \tag{12}$$

$$= P[\boldsymbol{u}^\top \boldsymbol{a}(\mathbf{X}_b) > 0]$$

$$= P[f_-(\mathbf{X}_b) > f_+(\mathbf{X}_b)]$$

$$\leq 1 - \psi \tag{13}$$

This is to say that when $\alpha = 1$, the classifier is not modified at all, thus the SIA will be no larger than $1 - \psi$ since the attack is $\psi$-successful (see Definition 3.2). On the other hand, by setting $\alpha = 0$, we will have the following:

$$
\begin{aligned}
P[g_-(\mathbf{X}_b|\alpha) > g_+(\mathbf{X}_b|\alpha)\big|\alpha = 0] &= P[\boldsymbol{u}^\top \hat{\boldsymbol{a}}(\mathbf{X}_b|\alpha) > 0\big|\alpha = 0] \\
&= P[\boldsymbol{u}^\top \boldsymbol{a}^*(\mathbf{X}_b) > 0] \\
&= P[f_-(\mathbf{X}_-) > f_+(\mathbf{X}_-)] \qquad\qquad (14) \\
&\geq 1 - \eta \qquad\qquad\qquad\qquad\qquad\;\; (15)
\end{aligned}
$$

and this is to say that when $\alpha = 0$, the distribution shift will be fully recovered, such that SIA is equally high as the accuracy of the source class. Recall that Eq. (14) is due to Eq. (7) and Eq. (8). The inequality (15) is because the classifier specified by $f_-$ and $f_+$ is assumed $\eta$-erroneous (see Definition 3.1). Here, we prove the theorem by showing that the partial derivative of $P[g_-(\mathbf{X}_b|\alpha) > g_+(\mathbf{X}_b|\alpha)]$ over $\alpha$ is strictly *negative* when $\sigma_b \leq \sigma$ (i.e. triggered instances have smaller standard deviation than clean instances, which is generally true). To achieve this, we notice that

$$
\boldsymbol{u}^\top \hat{\boldsymbol{a}}(\mathbf{X}_b|\alpha) = \boldsymbol{u}^\top \hat{\boldsymbol{V}}(\alpha)^{-\frac{1}{2}} \boldsymbol{\Gamma}(\boldsymbol{W}\mathbf{X}_b - \hat{\boldsymbol{m}}(\alpha)) + \boldsymbol{u}^\top \boldsymbol{\beta}
$$

follows a Gaussian distribution with

$$
\mathbb{E}[\boldsymbol{u}^\top \hat{\boldsymbol{a}}(\mathbf{X}_b|\alpha)] = \boldsymbol{u}^\top \hat{\boldsymbol{V}}(\alpha)^{-\frac{1}{2}} \boldsymbol{\Gamma}(-\boldsymbol{W}\boldsymbol{\mu} + \boldsymbol{W}\boldsymbol{\epsilon} - \hat{\boldsymbol{m}}(\alpha)) + \boldsymbol{u}^\top \boldsymbol{\beta} \qquad (16)
$$

$$
\mathrm{Var}[\boldsymbol{u}^\top \hat{\boldsymbol{a}}(\mathbf{X}_b|\alpha)] = \sigma_b^2 \|\boldsymbol{W}^\top \boldsymbol{\Gamma} \hat{\boldsymbol{V}}(\alpha)^{-\frac{1}{2}} \boldsymbol{u}\|_2^2 \qquad\qquad\qquad (17)
$$

We also notice that for source class instances $\mathbf{X}_-$

$$
\boldsymbol{u}^\top \boldsymbol{a}(\mathbf{X}_-) = \boldsymbol{u}^\top \boldsymbol{V}^{-\frac{1}{2}} \boldsymbol{\Gamma}(\boldsymbol{W}\mathbf{X}_- - \boldsymbol{m}) + \boldsymbol{u}^\top \boldsymbol{\beta}
$$

follows a Gaussian distribution with

$$
\mathbb{E}[\boldsymbol{u}^\top \boldsymbol{a}(\mathbf{X}_-)] = \boldsymbol{u}^\top \boldsymbol{V}^{-\frac{1}{2}} \boldsymbol{\Gamma}(-\boldsymbol{W}\boldsymbol{\mu} - \boldsymbol{m}) + \boldsymbol{u}^\top \boldsymbol{\beta} \qquad\qquad (18)
$$

$$
\mathrm{Var}[\boldsymbol{u}^\top \boldsymbol{a}(\mathbf{X}_-)] = \sigma^2 \|\boldsymbol{W}^\top \boldsymbol{\Gamma} \boldsymbol{V}^{-\frac{1}{2}} \boldsymbol{u}\|_2^2 \qquad\qquad\qquad (19)
$$

Then we have

$$
P[\boldsymbol{u}^\top \hat{\boldsymbol{a}}(\mathbf{X}_b|\alpha) > 0] = 1 - \boldsymbol{\Phi}\left(-\frac{\mathbb{E}[\boldsymbol{u}^\top \hat{\boldsymbol{a}}(\mathbf{X}_b|\alpha)]}{\sqrt{\mathrm{Var}[\boldsymbol{u}^\top \hat{\boldsymbol{a}}(\mathbf{X}_b|\alpha)]}}\right) \qquad (20)
$$

$$
P[\boldsymbol{u}^\top \boldsymbol{a}(\mathbf{X}_-) > 0] = 1 - \boldsymbol{\Phi}\left(-\frac{\mathbb{E}[\boldsymbol{u}^\top \boldsymbol{a}(\mathbf{X}_-)]}{\sqrt{\mathrm{Var}[\boldsymbol{u}^\top \boldsymbol{a}(\mathbf{X}_-)]}}\right) \qquad (21)
$$

where $\boldsymbol{\Phi}$ is the cumulative distribution function of standard Gaussian. Now let's consider Eq. (21) first. Since $\eta < \frac{1}{2}$ as we have reasonably assumed (otherwise the classifier may be worse than a random guess), and also according to Eq. (15), we have

$$
P[\boldsymbol{u}^\top \boldsymbol{a}(\mathbf{X}_-) > 0] = P[f_-(\mathbf{X}_-) > f_+(\mathbf{X}_-)] > \frac{1}{2}
$$

Thus, based on Eq. (21) and Eq. (18), we get

$$
\boldsymbol{u}^\top \boldsymbol{V}^{-\frac{1}{2}} \boldsymbol{\Gamma}(-\boldsymbol{W}\boldsymbol{\mu} - \boldsymbol{m}) + \boldsymbol{u}^\top \boldsymbol{\beta} > 0 \qquad\qquad (22)
$$

Next, let's focus on Eq. (20). Again, we set $\alpha = 1$. Based on (12)-(13) and the reasonable assumption that $\psi > \frac{1}{2}$ (otherwise the attack is not deemed successful since the success rate will be even lower than the accuracy on clean instances), we have

$$
P[\boldsymbol{u}^\top \hat{\boldsymbol{a}}(\mathbf{X}_b|\alpha) > 0\big|\alpha = 1] = P[f_-(\mathbf{X}_b) > f_+(\mathbf{X}_b)] < \frac{1}{2}
$$

Then, based on Eq. (20) and Eq. (16), we get

$$
\boldsymbol{u}^\top \boldsymbol{V}^{-\frac{1}{2}} \boldsymbol{\Gamma}(-\boldsymbol{W}\boldsymbol{\mu} + \boldsymbol{W}\boldsymbol{\epsilon} - \boldsymbol{m}) + \boldsymbol{u}^\top \boldsymbol{\beta} < 0 \qquad\qquad (23)
$$

Subtract Eq. (23) from Eq. (22) we get:

$$
-\boldsymbol{u}^\top \boldsymbol{V}^{-\frac{1}{2}} \boldsymbol{\Gamma} \boldsymbol{W}\boldsymbol{\epsilon} > 0 \qquad\qquad\qquad (24)
$$

Based on Eq. (20), we also have

$$\frac{\partial P[g_-(\mathbf{X}_b|\alpha) > g_+(\mathbf{X}_b|\alpha)]}{\partial \alpha} = \frac{\partial \frac{\mathbb{E}[\boldsymbol{u}^\top \hat{\boldsymbol{a}}(\mathbf{X}_b|\alpha)]}{\sqrt{\mathrm{Var}[\boldsymbol{u}^\top \hat{\boldsymbol{a}}(\mathbf{X}_b|\alpha)]}}}{\partial \alpha} \cdot \phi(-\frac{\mathbb{E}[\boldsymbol{u}^\top \hat{\boldsymbol{a}}(\mathbf{X}_b|\alpha)]}{\sqrt{\mathrm{Var}[\boldsymbol{u}^\top \hat{\boldsymbol{a}}(\mathbf{X}_b|\alpha)]}}) \qquad (25)$$

where $\phi$ is the probability density function (PDF) for standard normal distribution. Based on Eq. (16), Eq. (17), and Eq. (2), we have

$$\frac{\mathbb{E}[\boldsymbol{u}^\top \hat{\boldsymbol{a}}(\mathbf{X}_b|\alpha)]}{\sqrt{\mathrm{Var}[\boldsymbol{u}^\top \hat{\boldsymbol{a}}(\mathbf{X}_b|\alpha)]}}$$
$$= \frac{\boldsymbol{u}^\top \boldsymbol{V}^{-\frac{1}{2}}\boldsymbol{\Gamma}[-\boldsymbol{W}\boldsymbol{\mu} + \boldsymbol{W}\boldsymbol{\epsilon} - (\alpha \boldsymbol{m} + (1-\alpha)\boldsymbol{m}^*)] + (\alpha + (1-\alpha)\frac{\sigma_b}{\sigma})\boldsymbol{u}^\top \boldsymbol{\beta}}{\sigma_b ||\boldsymbol{W}^\top \boldsymbol{\Gamma} \boldsymbol{V}^{-\frac{1}{2}}\boldsymbol{u}||_2}$$
$$= \frac{\alpha \cdot \boldsymbol{u}^\top \boldsymbol{V}^{-\frac{1}{2}}\boldsymbol{\Gamma}[(\frac{\sigma_b}{\sigma} - 1)\boldsymbol{m} + (\frac{\sigma_b}{\sigma} - 1)\boldsymbol{W}\boldsymbol{\mu} + \boldsymbol{W}\boldsymbol{\epsilon}] - \alpha \cdot (\frac{\sigma_b}{\sigma} - 1)\boldsymbol{u}^\top \boldsymbol{\beta}}{\sigma_b ||\boldsymbol{W}^\top \boldsymbol{\Gamma} \boldsymbol{V}^{-\frac{1}{2}}\boldsymbol{u}||_2} + \text{constant}$$

and thus, based on Eq. (23) and Eq. (24)

$$\frac{\partial \frac{\mathbb{E}[\boldsymbol{u}^\top \hat{\boldsymbol{a}}(\mathbf{X}_b|\alpha)]}{\sqrt{\mathrm{Var}[\boldsymbol{u}^\top \hat{\boldsymbol{a}}(\mathbf{X}_b|\alpha)]}}}{\partial \alpha}$$
$$= \frac{(\frac{\sigma_b}{\sigma} - 1)[\boldsymbol{u}^\top \boldsymbol{V}^{-\frac{1}{2}}\boldsymbol{\Gamma}(\boldsymbol{m} + \boldsymbol{W}\boldsymbol{\mu} - \boldsymbol{W}\boldsymbol{\epsilon}) - \boldsymbol{u}^\top \boldsymbol{\beta}] + \frac{\sigma_b}{\sigma}\boldsymbol{u}^\top \boldsymbol{V}^{-\frac{1}{2}}\boldsymbol{\Gamma}\boldsymbol{W}\boldsymbol{\epsilon}}{\sigma_b ||\boldsymbol{W}^\top \boldsymbol{\Gamma} \boldsymbol{V}^{-\frac{1}{2}}\boldsymbol{u}||_2}$$
$$< 0$$

when $\sigma_b \leq \sigma$. Substitute it into Eq. (25) and given Gaussian PDF being strictly positive, we have

$$\frac{\partial P[g_-(\mathbf{X}_b|\alpha) > g_+(\mathbf{X}_b|\alpha)]}{\partial \alpha} < 0$$

$\square$

# B  DATASETS, TRAINING SETTINGS, AND ATTACK SETTINGS

## B.1  DATASETS

In experiments, we show the effectiveness of our proposed backdoor mitigation method on several benchmark datasets including CIFAR-10 (Krizhevsky & Hinton (2009)), GTSRB (Houben et al. (2013)), CIFAR-100 (Krizhevsky & Hinton (2009)), ImageNette (Howard (2020)), and TinyImageNet. CIFAR-10 dataset contains 60,000 $32 \times 32$ color images from 10 classes, with 5,000 images per class for training and 1,000 images per class for testing . GTSRB dataset has more than 50,000 traffic sign images with different sizes from 43 classes. Here, we resize all images in GTSRB to $32 \times 32$. CIFAR-100 contains 60,000 $32 \times 32$ color images evenly from 100 classes, where 500 images per class are used for training, while the others are used for testing. ImageNette is a subset of 10 easily classified classes from Imagenet[4], with image size of $256 \times 256$. For each class, there are around 900 images for training and 400 images for testing. The TinyImageNet dataset is a subset of the ImageNet dataset (Russakovsky et al. (2015)). It contains 100,000 $64 \times 64$ color images evenly distributed in 200 classes (500 training images and 50 test images for each class).

## B.2  TRAINING SETTINGS

Training settings for the 5 datasets are shown in Table 6. We train a ResNet-18 (He et al. (2016)) on CIFAR-10 and CIFAR-100 for 30 epochs and 40 epochs, respectively. We train a ResNet-34 (He et al. (2016)) on both TinyImageNet and ImageNette for 90 epochs. For GTSRB, we train a MobileNet (Howard et al. (2017)) for 60 epochs. For all models, we use Adam optimizer (Kingma & Ba (2015)) for parameter learning and a scheduler to decay the learning rate of each parameter group by 0.1 every "scheduler step size" epochs (shown in the table). We choose batch size 32 for both CIFAR-10 and CIFAR-100, 64 for GTSRB and ImageNette, and 128 for TinyImageNet.

## B.3  ATTACK SETTINGS

On dataset CIFAR-10, we consider the following triggers: 1) a $3 \times 3$ random patch (**BadNet**) with a randomly selected location (fixed for all triggered images for each attack) used in Gu et al. (2019), as visualized in Fig. 3b; 2) an additive perturbation (with size 2/255) resembling a chessboard (**CB**) used in Xiang et al. (2020), as visualized in Fig. 3c; 3) a single pixel (**SP**) perturbed by 75/255 with a randomly selected location (fixed for all triggered images for each attack) used by Tran et al. (2018), as visualized in Fig. 3d; 4) invisible triggers generated with $l_0$ and $l_2$ norm constraints ($l_0$ **inv** and $l_2$ **inv** respectively) proposed by Li et al. (2021a), as visualized in Fig. 3e and 3f; 5) a warping-based trigger (**WaNet**) proposed by Nguyen & Tran (2021), as visualized in Fig. 3g.

Attack settings for CIFAR-10 are summarized in Tab. 7. For all-to-one attacks, we arbitrarily choose class 9 as the target class, and embed the backdoor triggers in 100 randomly chosen training samples per class (excluding the target class). To achieve similar effective attacks as other triggers, we poison 900 images per source class in the all-to-one attack using WaNet. For all-to-all attacks, we embed the backdoor triggers into 300 images for each class. For effective attacks, we poison 800 training images and 1500 training images in the all-to-all attacks using SP and WaNet, respectively.

Attack settings for other datasets are summerized in Tab. 8. Due to the insufficiency of data, we only conduct all-to-one attacks on these datasets for effective attacks. We arbitrarily choose class 0 as the target class for CIFAR-100, GTSRB, and TinyImageNet, and class 9 for ImageNette. The classes other than the target class are all source classes. For CIFAR-100, we use the same BadNet, $l_0$ inv, and $l_2$ inv triggers as CIFAR-10. We increase the perturbation size to 6/255 for CB pattern for a effective backdoor attack. For each of the attack, we poison 10 images per source class using the above triggers. Trigger SP and WaNet are not considered since we can not launch a successful backdoor attack using the trigger on CIFAR-100. For GTSRB, in addition to the same triggers as CIFAR-100, 2e also use the warping-based trigger (WaNet). We poison 2% of the training images per source class using BadNet trigger and $l_2$ inv trigger, and 5% of the training images per source class with CB trigger and $l_0$ inv trigger. To achieve similar effective attacks, we embed WaNet trigger into 24% of the training images per source class. For TinyImageNet and ImageNette, we

---

[4]The 10 classes are tench, English springer, cassette player, chain saw, church, French horn, garbage truck, gas pump, golf ball, and parachute.

only consider BadNet as the trigger, as the DNN can not learn the backdoor mapping using the other (relatively simple and small) triggers in datasets that are much more complicated than CIFAR-10. To successfully plant backdoors, we increase the size the the BadNet patch to $6 \times 6$ for TinyImageNet and to $21 \times 21$ for ImageNette. We embed the trigger in 10 training images per source class in TinyImageNet and in 5% of the training images per source class for ImageNette.

| Dataset | CIFAR-10 | CIFAR-100 | TinyImageNet | ImageNette | GTSRB |
|---|---|---|---|---|---|
| DNN architecture | ResNet-18 | ResNet-18 | ResNet-34 | ResNet-34 | MobileNet |
| Optimizer | Adam | Adam | Adam | Adam | Adam |
| Batch size | 32 | 32 | 128 | 64 | 64 |
| Epochs | 30 | 40 | 90 | 90 | 60 |
| Initial learning rate | 1e-3 | 1e-3 | 1e-3 | 1e-3 | 1e-3 |
| scheduler step size | 10 | 10 | 30 | 30 | 20 |

Table 6: Training configurations of the 5 datasets used in our experiments.

| Trigger type | BadNet | | CB | | $l_0$ inv | |
|---|---|---|---|---|---|---|
| | A2O | A2A | A2O | A2A | A2O | A2A |
| # poisoned per class | 100 | 300 | 100 | 300 | 100 | 300 |
| $l_0$ norm | $3 \times 3$ | $3 \times 3$ | NA | NA | $1 \times 6$ | $1 \times 6$ |
| $l_2$ norm | NA | NA | 0.3074 | 0.3074 | NA | NA |
| Trigger type | $l_2$ inv | | SP | | WaNet | |
| | A2O | A2A | A2O | A2A | A2O | A2A |
| # poisoned per class | 100 | 300 | 100 | 800 | 900 | 1500 |
| $l_0$ norm | NA | NA | NA | NA | NA | NA |
| $l_2$ norm | 1.6106 | 1.6106 | 0.5094 | 0.5094 | NA | NA |

Table 7: Attack configurations on CIFAR-10

| Trigger type | CIFAR-100 | | | | TinyImageNet | ImageNette |
|---|---|---|---|---|---|---|
| | BadNet | CB | $l_0$ inv | $l_2$ inv | BadNet | BadNet |
| Target class | 0 | 0 | 0 | 0 | 0 | 9 |
| # poisoned per class | 10 | 10 | 10 | 10 | 10 | 5% |
| $l_0$ norm | $3 \times 3$ | NA | $1 \times 6$ | NA | $6 \times 6$ | $21 \times 21$ |
| $l_2$ norm | NA | 0.9222 | NA | 1.6106 | NA | NA |
| Trigger type | GTSRB | | | | | |
| | BadNet | CB | $l_0$ inv | $l_2$ inv | WaNet | |
| Target class | 0 | 0 | 0 | 0 | 0 | |
| # poisoned per class | 2% | 5% | 5% | 2% | 24% | |
| $l_0$ norm | $3 \times 3$ | NA | $1 \times 6$ | NA | NA | |
| $l_2$ norm | NA | 0.9222 | NA | 1.6106 | NA | |

Table 8: Attack configurations on GTSRB, CIFAR-100, ImageNette, and TinyImageNet.

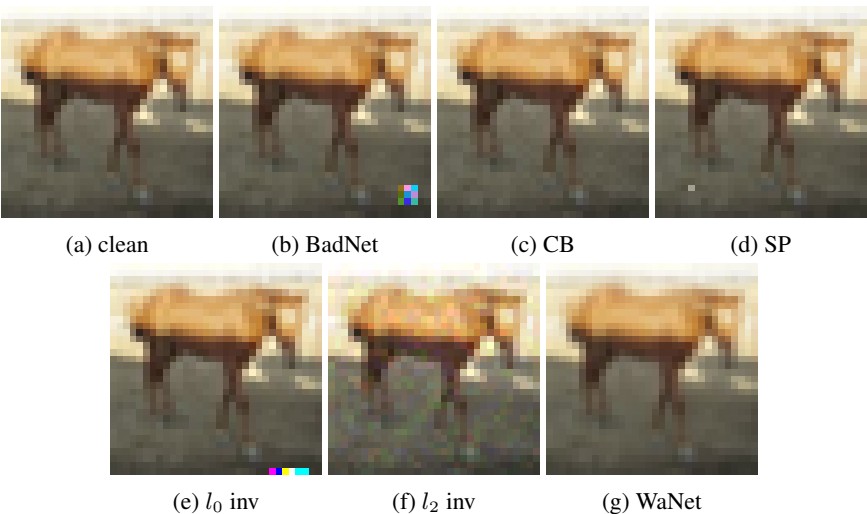

(a) clean (b) BadNet (c) CB (d) SP

(e) $l_0$ inv (f) $l_2$ inv (g) WaNet

Figure 3: Example of CIFAR-10 images embedded with the backdoor triggers considered in our experiments.

## C  PATTERN ESTIMATION AND BACKDOOR DETECTION

For our BNA, following Sec. 4.2, we first perform detection by reverse-engineering a backdoor trigger for each class pair. For patch triggers like BadNet, we use the objective function from Wang et al. (2019) for trigger reverse-engineering. For other more subtle, perturbation-based trigger types, we use the objective function from Xiang et al. (2020) for reverse-engineering. The detection statistic is the reciprocal of the $l_0$ norm of estimated patch triggers and $l_2$ norm of reverse-engineered perturbation-based triggers. Then we feed the statistics obtained from the estimated trigger to an anomaly detector.

Our anomaly detector is based on MAD, which is a classical approach also used by Wang et al. (2019); Chen et al. (2019b); Wang et al. (2020). It first calculates absolute deviation between all detection statistics (the reciprocal of $l_0$ norm of patch triggers and $l_2$ norm of perturbation-based triggers) and the median, and the median of the absolute deviations is called Median Absolute Deviation (MAD). For a class pair and its corresponding estimated trigger, if the trigger's anomaly score, which is defined as the absolute deviation divided by MAD, is larger than a given threshold, it is detected as a backdoor class pair. The detection threshold can be easily found, as shown in the Fig. 4 and Fig. 5. Fig. 4 and Fig. 5 show the histograms of the anomaly scores for all class pairs under all-to-one and all-to-all attacks, respectively. Here, we set the detection threshold at 7, which easily catches all the backdoor class pairs under all the attacks, except for the all-to-all BadNet attack and both attacks using WaNet trigger.

For the all-to-all BadNet attack, the outlier detector finds two source classes – 0 and 8 – for the target class 1, where 0-1 is the true source-target class pair and 8-1 is falsely detected, as shown in Fig. 5a. The $l_0$ norm of the trigger estimated on class 0 clean images is 3.02, and that estimated on class 8 images is 7.95. If class 0 and 8 are both the source classes involved in the backdoor attack, then the trigger estimated on the clean images from class 0 *and* 8 should also have a small $l_0$ norm. Otherwise, the trigger estimated using class 8 images is an intrinsic backdoor pattern (Xiang et al. (2022); Liu et al. (2022); Tao et al. (2022)), and 0-1 is the true source class pair, since the trigger of 0-1 has smaller size than 8-1. By optimizing on clean images from class 0 *and* 8, the $l_0$ norm of the trigger that causes mis-classification to class 1 with high confidence is 27.18 – much larger than the triggers estimated on either class 0 images or class 8 images. Thus, we detect 0-1 as the true backdoor class pair and discard the trigger for class pair 8-1 in backdoor mitigation.

For the attacks using warping-based triggers (WaNet), unlike the other attacks, trigger size for clean class pairs and backdoor class pairs are both small. However, there is still a "gap" between the anomaly scores of clean class pairs and backdoor class pairs, as shown in Fig. 4f and 5f. The outlier detector successfully detects all the backdoor class pairs by using a threshold at 3.

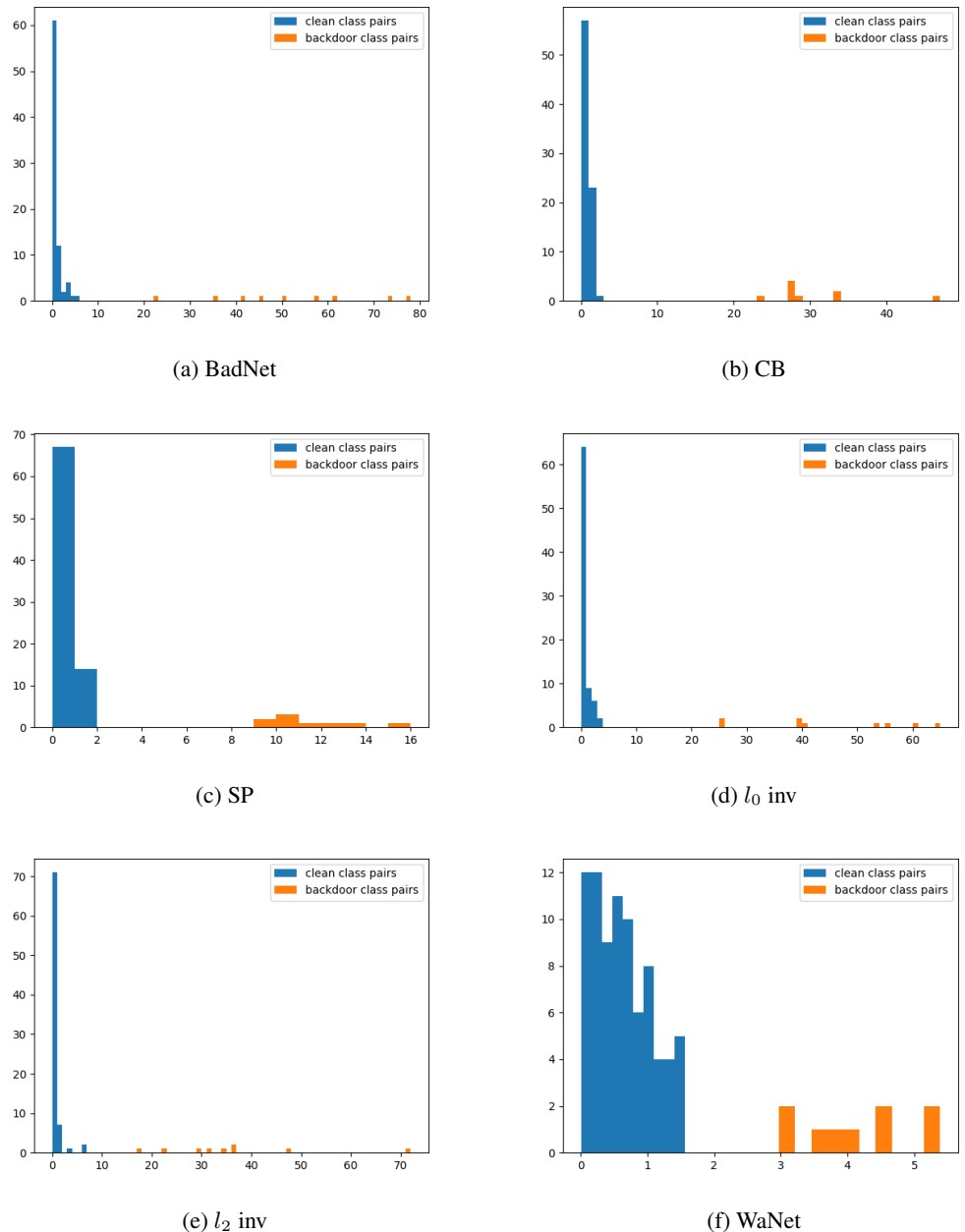

Figure 4: Histograms of anomaly scores for each class pair under all all-to-one attacks.

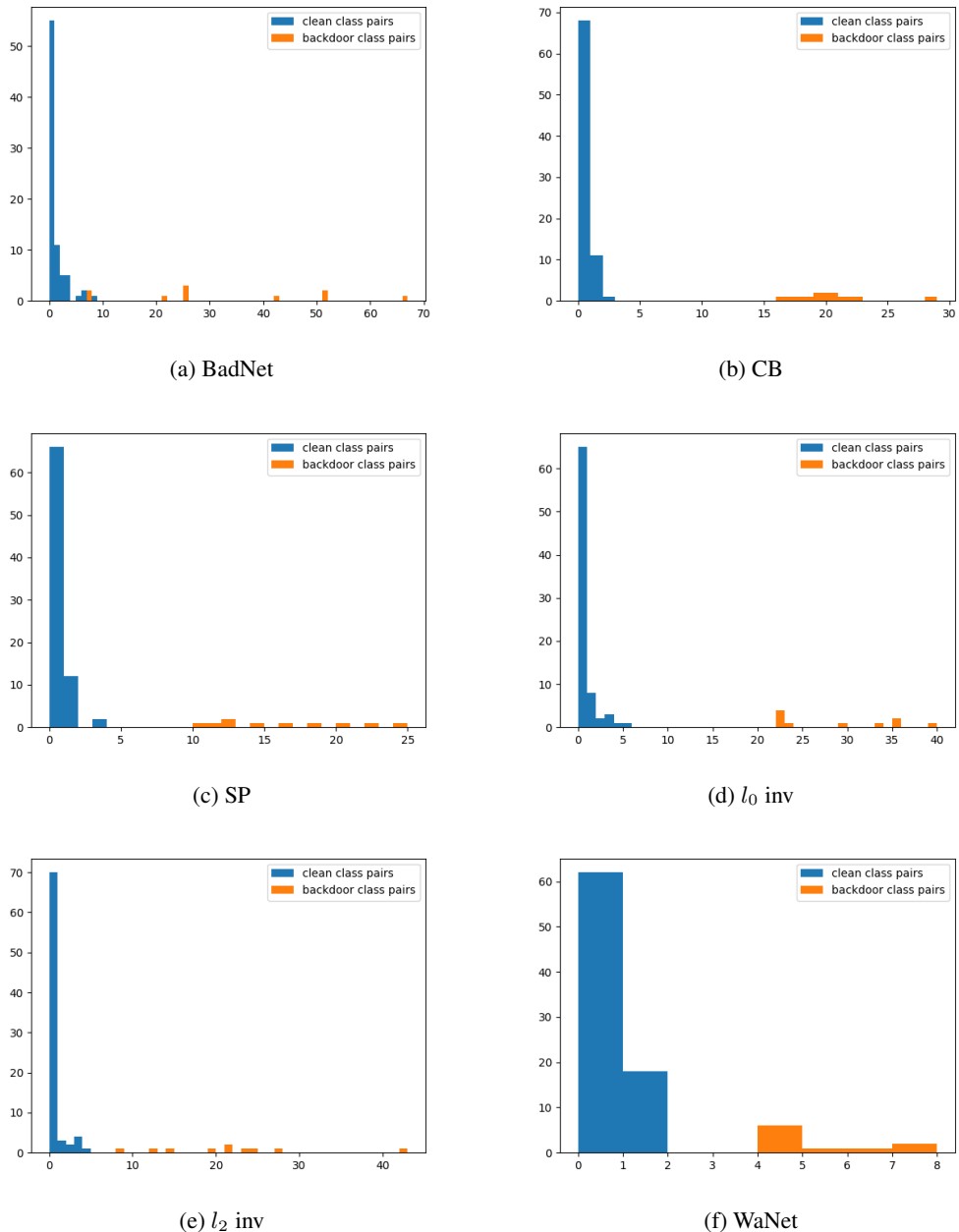

Figure 5: Histograms of anomaly scores for each class pair under all all-to-all attacks.

## D  DISTRIBUTION DIVERGENCES

As stated in Thm. 3.1, for our backdoor mitigation method, the SIA monotonically increases as the divergence between clean instances and backdoor-trigger instances decreases. We show the ACC, ASR, and SIA for our method against all all-to-one attacks on CIFAR-10 dataset in Tab. 1. Here, we show the corresponding distribution divergences under all attacks in Tab. 9. Tab. 9 shows the average TV distance, JS divergence, and KL divergence between distributions of penultimate layer activations of clean images and backdoor-trigger images in clean ResNet-18, backdoor poisoned ResNet-18, and backdoor poisoned ResNet-18 with BNA. For the backdoor poisoned ResNet-18 with BNA, we use TV divergence in backdoor mitigation. For all attacks, all the three divergences are small for a clean DNN, while relatively large for a backdoor-poisoned DNN. The distribution of backdoor-trigger instances severely deviates from that of clean instances. However, with our mitigation method, the distribution alteration is significantly relieved. All the three divergences are drastically reduced, which is consistent with the results in Tab. 1.

| Trigger type | BadNet | CB | $l_0$ inv | $l_2$ inv | SP | WaNet |
|---|---|---|---|---|---|---|
| | | | KL divergence | | | |
| clean | 0.0022 | 0.001 | 0.0013 | 0.0085 | 0.0017 | 0.0037 |
| poisoned | 0.3211 | 0.752 | 0.4029 | 0.873 | 0.4385 | 0.2708 |
| BNA | 0.0291 | 0.0141 | 0.0337 | 0.0402 | 0.0146 | 0.0389 |
| | | | JS divergence | | | |
| clean | 0.0239 | 0.0166 | 0.0185 | 0.0461 | 0.0214 | 0.0311 |
| poisoned | 0.2867 | 0.3528 | 0.3275 | 0.3898 | 0.3041 | 0.2215 |
| BNA | 0.0952 | 0.0583 | 0.0847 | 0.0986 | 0.0582 | 0.0611 |
| | | | TV distance | | | |
| clean | 634.1953 | 376.4219 | 461.9531 | 1240.3359 | 554.3594 | 805.9648 |
| poisoned | 8877.2734 | 11177.3867 | 9203.9609 | 12482.4336 | 10119.3281 | 7237.7812 |
| BNA | 2463.582 | 1705.0156 | 2490.457 | 2794.7578 | 1761.2148 | 1749.2461 |

Table 9: Average TV distance, JS divergence, and KL divergence between distributions of clean instances and backdoor-trigger instances in clean DNN, poisoned DNN, and poisoned DNN with BNA using TV divergence in backdoor mitigation.

## E  CHOICE OF DIVERGENCE FORMS

In Tab. 1 and 3, we only show the results for our method of using TV distance in backdoor mitigation (Eq. 3). Here we show that our method is not sensitive to the choice of distribution divergence form. We respectively use TV distance, JS divergence, and KL divergence to mitigate the 5 all-to-one CB attack against CIFAR-10, and show the average distribution similarity measured by the three measurements after mitigation. The distribution similarity is calculated on the penultimate layer activations. As shown in Tab. 10, the distribution alteration is significantly relieved after mitigation, regardless the measurement used in mitigation (Eq. 3).

| divergence form used in mitigation → 
 distribution similarity after mitigation ↓ | TV | JS | KL |
|---|---|---|---|
| TV | 1742 | 1730 | 1719 |
| JS | 0.0622 | 0.06172 | 0.0613 |
| KL | 0.0165 | 0.0163 | 0.0161 |

Table 10: Average TV, JS, and KL between clean instances and backdoor-trigger instances using TV, JS, and KL for measuring distribution similarity in backdoor mitigation.

## F  IMPACT OF PERTURBATION SIZE AND POISONING RATIO ON BACKDOOR MITIGATION

To observe the impact of attack settings on the performance of backdoor mitigation methods, we tune the poisoning ratio (*i.e.*, the number of poisoned instances per source class) and perturbation size used in all-to-one CB attacks, and apply all the mitigation methods on these poisoned DNNs. The results are shown in Tab. 11. Generally, the metrics for all methods decrease with increasing poisoning ratio and perturbation size. Although the performance for our BNA slightly declines as the attack is strengthened, our method still outperforms other methods in terms of the SIA. Besides, it achieves the best or comparable ACC and ASR to other methods.

| Mitigation method | | the number of poisoned instances per class | | | | | perturbation size (*255) | | | | |
|---|---|---|---|---|---|---|---|---|---|---|---|
| | | 50 | 100 | 150 | 200 | 250 | 2 | 3 | 4 | 5 | 6 |
| NC | ACC | 0.8953 | 0.8734 | 0.8826 | 0.8943 | 0.8799 | 0.8734 | 0.8918 | 0.8667 | 0.8825 | 0.8709 |
| | ASR | **0.0056** | 0.0238 | 0.0148 | **0.0057** | 0.0064 | 0.0238 | **0.0042** | 0.0157 | 0.0133 | **0.0065** |
| | SIA | 0.8515 | 0.8412 | 0.8552 | 0.8579 | 0.8532 | 0.8412 | 0.8560 | 0.8283 | 0.8194 | 0.7883 |
| I-BAU | ACC | 0.8708 | 0.8473 | 0.8818 | 0.8941 | 0.8594 | 0.8473 | 0.8646 | 0.8822 | **0.9004** | 0.8919 |
| | ASR | 0.0789 | **0.0043** | 0.0712 | 0.5074 | 0.0460 | **0.0043** | 0.3052 | 0.0247 | **0.0011** | 0.2048 |
| | SIA | 0.6564 | 0.8399 | 0.7563 | 0.3802 | 0.6715 | 0.8399 | 0.5823 | 0.7712 | 0.8102 | 0.5153 |
| ANP | ACC | 0.8523 | 0.8271 | 0.8612 | 0.8204 | 0.7614 | 0.8271 | 0.8486 | 0.8156 | 0.8418 | 0.8249 |
| | ASR | 0.1940 | 0.8535 | 0.5401 | 0.0031 | **0.0043** | 0.8535 | 0.9836 | 0.6394 | 0.3670 | 0.2548 |
| | SIA | 0.5158 | 0.1047 | 0.2440 | 0.6157 | 0.3701 | 0.1047 | 0.0142 | 0.1911 | 0.3238 | 0.4606 |
| NAD | ACC | 0.8942 | 0.8745 | 0.8949 | 0.8902 | 0.8674 | 0.8767 | 0.8823 | 0.8745 | 0.8835 | **0.8990** |
| | ASR | 0.0147 | 0.0086 | **0.0095** | 0.0106 | 0.0125 | 0.0070 | 0.0096 | 0.0086 | 0.0642 | 0.0586 |
| | SIA | 0.8646 | 0.8504 | 0.8709 | 0.8695 | 0.8514 | 0.8631 | 0.8574 | 0.8504 | 0.8070 | 0.7896 |
| ARGD | ACC | 0.8743 | 0.8832 | 0.8693 | 0.8659 | 0.8415 | 0.8832 | 0.8872 | 0.8619 | 0.8508 | 0.8394 |
| | ASR | 0.0106 | 0.0108 | 0.0097 | 0.0117 | 0.0121 | 0.0108 | 0.0073 | **0.0083** | 0.0085 | 0.0153 |
| | SIA | 0.8590 | 0.8685 | 0.8528 | 0.8482 | 0.8267 | 0.8685 | 0.8574 | 0.8467 | 0.8373 | 0.8196 |
| BNA (ours) | ACC | **0.9112** | **0.9094** | **0.9098** | **0.9102** | **0.9015** | **0.9094** | **0.9041** | **0.9079** | 0.8992 | 0.8912 |
| | ASR | 0.0095 | 0.0141 | 0.0121 | 0.0170 | 0.0090 | 0.0141 | 0.0395 | 0.0222 | 0.0109 | 0.0388 |
| | SIA | **0.8851** | **0.8837** | **0.8728** | **0.8840** | **0.8662** | **0.8851** | **0.8783** | **0.8711** | **0.8814** | **0.8435** |

Table 11: ACC, ASR, and SIA for our BNA, NC, I-BAU, ANP, NAD, and ARGD as a function of (1) the number of poisoned instances injected into the training set; (2) the perturbation size under all-to-one CB attack.

