# OpenReview forum: "Backdoor Mitigation by Correcting Activation Distribution Alteration"
_ICLR.cc/2023/Conference — Submitted to ICLR 2023_

### Official Review · Reviewer_y2Ez · 2022-10-17

**Confidence:** 4
**Clarity, Quality, Novelty And Reproducibility:** good
**Correctness:** 3
**Technical Novelty And Significance:** 2
**Empirical Novelty And Significance:** Not applicable
**Recommendation:** 3

**Strength And Weaknesses:**

Strengths:

- Trendy topic

Weaknesses:

- Lack of novelty
- Hard-to-follow
- More explanations needed

Comments for the authors:

Defense algorithm against DNNs backdoor attack has been a popular topic. In this paper, the authors propose a post-training backdoor mitigation framework to detect triggered inputs and output their original classes. The main intuition of the proposed framework is using a property of backdoor attack: the triggered instances could cause an alteration in the distribution of internal layer activations. Therefore, the authors leverage reverse-engineered triggers to correct  this distribution. The advantages of the proposed framework are twofold: Non-interference in the training process of the model and low computational complexity.

However, I do have the following concerns.

- First, the main contribution of this paper needs to be stated more clearly. The authors claim that they discover the triggered inputs can result in distribution alteration.  However, a similar conclusion has been proposed by Tran et al. (2018). Meanwhile, the authors state that the detector in the framework is different from most existing works, so what is the difference between this paper and Wang et al. (2019) and Xiang et al. (2020).

- The writing of this paper needs to be improved. For example, this paper uses lots of notations which makes it hard to follow. The presentation about the choice of the parameters and the construction of the transformation function should be more clear. The authors could simplify them by using simple notations or list them in a table.

- In the defender’s assumptions, the authors assume that the defender has no access to the training set of the classifier, but does possess an independent clean dataset. What’s the meaning of "independent" here? What are the distribution requirements for this clean dataset? In the experiments, the authors leverage 10% samples from the test set, which has the same distribution as the training set. However, we can acquire a classifier from the wild and we cannot know its training dataset. So, how can we choose this clean dataset in such a case?

- More experiments are needed to demonstrate the generality of the proposed framework. For instance, the theoretical analysis focuses on the DNNs with Batch Normalization (BN) layers. However, the DNNs could leverage Group Normalization (GN) instead of BN. In the experiments, the architectures of DNNs are ResNet and MobileNet (both using BN). I would appreciate it if the authors could consider more different architectures.

- Recently, Jia et al. [1] proposed a novel entangled backdoor attack. This attack leverages the soft nearest neighbor loss to entangle representations extracted from clean data and triggered data, which allows the model to use the same subset of parameters to recognize clean data and malicious data. As a result, the divergence of activation distribution between clean and backdoored instances is small, so I am curious whether the proposed method can detect such entangled attacks.

[1] Jia H, Choquette-Choo C A, Chandrasekaran V, et al. Entangled watermarks as a defense against model extraction[C]//30th USENIX Security Symposium (USENIX Security 21). 2021: 1937-1954.


**Summary Of The Paper:**

Paper Summary: Deep neural networks (DNNs) are vulnerable to backdoor attacks, and this paper proposes a mitigation framework to detect malicious instances and classify the triggered instance to its original class in the inference time.


**Summary Of The Review:**

see strength and weakness

---

> ### Author Response · Authors · 2022-11-14
> **Clarification for novelty and assumption for the defenders**
>
> We thank the reviewers for providing valuable feedback to our work. We will improve the paper based on the reviews. Below we clarify misunderstandings and address questions of the reviewer.
> 1. Thanks for the comment -- we will add the following clarification to our paper to avoid similar misunderstanding. First, the observation made by Tran et al. (2018) was mentioned in our introduction, and is *entirely different* from our observations. Using the words in our paper, they observed that “triggered instances (labeled to the target class) are separable from clean *target* class instances in terms of internal layer activations of the poisoned classifier”. They leverage this property to design a *pre-training* outlier detection method, where they detect and remove triggered instances from the poisoned training set, then re-train the DNN on the remaining samples. Differently, the distribution alteration in our property 3.1 resides between the triggered instances and the clean *source* class instances (the source class is the original class for the triggered class). Based on this property, we designed a *post-training* mitigation method (as we emphasized in the title and paper), where we do not have access to the training set. After mitigation with a small dataset, the model will correctly classify triggered instances to their original source classes at test-time. Second, our paper focuses on backdoor mitigation (not detection), which is the fundamental difference from Wang et al. (2019) and Xiang et al. (2020). These two papers focused on backdoor detection, where the detector infers whether a given model is attacked or not. The detection methods cannot be used as mitigation methods, since the model that is detected to be backdoor poisoned will still classify to the target class when the trigger is part of the input (as we mentioned in Sec. 2). We applied these two detection methods before mitigation to estimate backdoor triggers, and compared our method with other *mitigation* methods (as shown in Sec. 5.2).
> 2. Thanks for this comment. We will improve the writing, add a table of notations in the appendix, and add a pseudocode for our method.
> 3. “Independent” means that these sample are collected by the defender but are not used during training. That is, this small clean dataset does not overlap with the training set. We only need the clean samples to come from the same classification domain as the original training set (or the training set used for model transfer learning). There is no distribution requirement for this clean dataset. This is a common assumption used by many post-training backdoor detection and mitigation methods, e.g., Wang et al. (2019); Xiang et al. (2020); Wang et al. (2020); Liu et al. (2019); Liu et al. (2018a); Wu & Wang (2021); Guan et al. (2022); Zheng et al. (2022)); Li et al. (2021c); Xia et al. (2022); and Zeng et al. (2022). For the benchmark datasets used in our experiments, there is no guarantee that the test set has the same distribution as the training set. Besides, the test accuracy for these datasets is uniformly lower than the training accuracy. Moreover, if these test examples can be used for validating the classifier’s accuracy, they should also be valid for backdoor detection etc. In practice, the user/defender does not need to know its training set. It only needs to know the classification domain, which is natural. Then it can collect clean samples for detection/mitigation independently.
> 4. In our paper, the observation on distribution alteration and the distribution alignment method do not rely on the existence of BN layers (please see Eq (4)). We included BN in our analysis just for generalization. One could simply set gamma=1 and beta=0 when there is no BN; and the analysis still holds. Nevertheless, we will include an experiment in our revision for an architecture without BN.
> 5. Thanks for pointing this out. Actually, in Sec. 5.4, we have an adaptive attack which is similar to the attack [1], in terms of the goal to reduce the distance between instances with and without the trigger in the internal layer. Nevertheless, we will consider [1] and show the mitigation results in our experiments.

---

### Official Review · Reviewer_zrrB · 2022-10-23

**Confidence:** 4
**Correctness:** 2
**Technical Novelty And Significance:** 3
**Empirical Novelty And Significance:** 2
**Recommendation:** 5

**Clarity, Quality, Novelty And Reproducibility:**

* This paper tends to use mathematical equations to explain ideas and methods, which is good. However, it fails to provide intuitive introductions to the points/ideas that it aims to elaborate. Section 3 is somewhat easy to follow as most parts have intuitive explanations before diving into detailed math equations. It however becomes hard to understand in section 4.2, especially starting from page 6. There are a lot of details without giving an overview of what this paper aims to address. Intuitively, it seems trying to explain what the transformation function is and how it is estimated. But it is really hard to grasp the main procedure without a clear outline. It is suggested to rewrite the whole section 4.2 to make it more structured and intuitive. Details shall be explained after the outline is clearly established.

* Theorem 3.1 is based on the assumption that \sigma_b \leq \sigma. This paper uses a patch attack and additive attacks as examples to validate this assumption. However, it lacks empirical results to validate this assumption. Particularly, there are many complex backdoor attacks such as DFST [1], reflection backdoor [2], composite backdoor [3]. It is very likely they violate the assumption. This paper shall empirically study them. In addition, this paper only considers universal attack, where samples from all classes are misclassified to a target class. It does not consider label-specific attack, which only causes misclassification for one particular source class. Such an attack may not reduce the variance. This paper shall study this type of attacks or state clearly in their threat model.

* The first part of the proposed method just uses existing trigger inversion techniques. However, it does not clearly explain how trigger inversion works, which makes the paper not self-contained. Since challenges 1&2 are directly solved by existing techniques, it does not seem reasonable to claim this part in the proposed method. It would be more clear to separate the first two challenges and the last one using subsections.

* Some results in Table 1 do not seem reasonable. For example, BadNet and WaNet can easily achieve near 100% attack success rate. However, they are much lower in the table. This paper uses generated pair-wise triggers to eliminate injected backdoors, which is similar to existing method [4]. It shall also be compared with in the evaluation.

* As mentioned earlier, there are many complex backdoor attacks [1-3], which may violate the assumption that the proposed defense is based on. It is suggested to evaluate on those attacks to see the performance of the proposed defense.


### References

[1] Cheng, Siyuan, et al. "Deep feature space trojan attack of neural networks by controlled detoxification." AAAI 2021.

[2] Liu, Yunfei, et al. "Reflection backdoor: A natural backdoor attack on deep neural networks." ECCV 2020.

[3] Lin, Junyu, et al. "Composite backdoor attack for deep neural network by mixing existing benign features." CCS 2020.

[4] Tao, Guanhong, et al. "Model orthogonalization: Class distance hardening in neural networks for better security." IEEE S&P 2022.

**Strength And Weaknesses:**

Strength

+ Important topic of backdoor removal
+ Interesting perspective of adjusting activation values

Weaknesses

- Many parts of the method are not clearly explained
- Strong assumption of backdoor attacks and the theorem may not hold
- Lack of background for trigger inversion
- No comparison with state-of-the-art backdoor removal techniques
- No evaluation on recent and complex backdoor attacks

**Summary Of The Paper:**

This paper observes that the activation distributions are different for clean inputs and backdoor samples. It hence proposes to change activation values of inputs such that the predictions of backdoor samples can be corrected. Particularly, this paper utilizes existing trigger inversion methods to find a set of suspicious targets and generate corresponding backdoor triggers. For inputs with predicted labels in the set, it adjusts the activation values by using a transformation function. The transformation function changes the mean and standard deviation of activation values, which is obtained by comparing the activation differences between a set of clean samples with and without the inverted trigger. The experiment is conducted on five datasets and six types of backdoor attacks. Compared to several backdoor removal methods, this proposed approach can achieve good defense performance.

**Summary Of The Review:**

The idea of transforming activation values to remove backdoors is interesting and the results show potential. However, the paper lacks structure and is hard to understand. In addition, more empirical studies are needed to further validate the defense assumption and the performance on more complex backdoor attacks.

---

> ### Author Response · Authors · 2022-11-14
> **Clarification for experiment settings**
>
> We thank the reviewers for providing valuable feedback to our work. We will improve the paper based on the reviews. Below we clarify misunderstandings and address questions of the reviewer.
> 1. Thanks for pointing this out. We will rewrite Sec. 4.2 with pseudocode.
> 2. We considered all-to-all attacks, where we have one source class for one target class (Please see the detailed attack settings in Sec. 5.1). We think such an attack setting is a label-specific attack. Besides, apart from patch attacks and additive attacks, we considered WaNet[5], which is one of the sample-specific backdoor attacks. We will consider the three attacks [1-3] in our experiments.
> 3. Thanks for pointing this out. We will add related background knowledge in Sec. 2. In fact, we used the existing trigger reverse-engineering techniques, but a different detection inference scheme to serve our goal of mitigation and test-time detection (which is described in Sec. 4.2).
> 4. Both attack settings and training settings will affect the ASR. We used the same settings for all attacks considered in our paper, which are shown in Sec. 5.1 and appendix B. Here we consider practical attack settings, where the attacker only has access to a few training samples, and the settings are different from the original ones. (WaNet assumes the attacker has access to all training samples, which is impossible in practice.) In our main table (table 1), we considered attacks with ASR>90% as effective attacks, and compared our method with other state-of-the-art methods under the same attacks. Then in table 2, we showed that our method performs stably with increasing attack strength (perturbation size and poisoning ratio), and compared with other methods under stronger attacks in table 11 in appendix F. Besides, for other datasets, we have ASR near 100% in some settings (please see table 3), and our method still effectively mitigates the attacks. In other words, our method works under *strong attacks*. [4] used a generated trigger for training to enlarge the distance between classes, such that existing natural backdoors (first observed by Xiang et at. in [6] and named “intrinsic backdoors”) will likely be eliminated, which increases the effectiveness of backdoor detection. Despite [4] using reverse-engineered triggers in a very different way than in our work, their approach does not mitigate backdoor attacks. We will cite this paper, but it is difficult to compare with our method since we are solving different problems.
>
> [1] Cheng, Siyuan, et al. "Deep feature space trojan attack of neural networks by controlled detoxification." AAAI 2021.
>
> [2] Liu, Yunfei, et al. "Reflection backdoor: A natural backdoor attack on deep neural networks." ECCV 2020.
>
> [3] Lin, Junyu, et al. "Composite backdoor attack for deep neural network by mixing existing benign features." CCS 2020.
>
> [4] Tao, Guanhong, et al. "Model orthogonalization: Class distance hardening in neural networks for better security." IEEE S&P 2022.
>
> [5] Tuan Anh Nguyen and Anh Tuan Tran. Wanet - imperceptible warping-based backdoor attack. In ICLR, 2021.
>
> [6] Zhen Xiang, David J. Miller, Siheng Chen, Xi Li, and George Kesidis. Detecting backdoor attacks against point cloud classifiers. In ICASSP, 2022

---

### Official Review · Reviewer_2vpF · 2022-10-28

**Confidence:** 4
**Correctness:** 3
**Technical Novelty And Significance:** 2
**Empirical Novelty And Significance:** 2
**Recommendation:** 3

**Clarity, Quality, Novelty And Reproducibility:**

The writing is generally not very clean and a bit hard to read.

Novelty is not strong as explained above.

**Strength And Weaknesses:**

The writing/presentation of the paper is not particularly well structured. A pseudocode can be helpful.

While the experiment does seem to support the claim, I have reservations in both novelty and motivation.

The observation that neural activations of clean and triggered samples at the penultimate layer have different distribution is not particularly surprising, given it is well known that the softmax output of clean and triggered samples are different. The idea of mitigating by pulling the distribution together also seems incremental.

A more serious issue is the fact that the method is highly dependent on the success of the existing detection method, and dependent on the reverse engineered triggers. This makes it hard to believe that the method is robust and generalizes well. First, as attacks continue evolving, it is hard to believe the used classic detection method is still as effective. Second, with so many engineering components involved, the method will be extremely hard to tune.

Motivation-wise it is also unclear: if a model is already detected to be attacked, why is it worthy to mitigate it anyway?  A convincingly good mitigation method should be easy and simple to use. It should work despite whether the model is attacked.

**Summary Of The Paper:**

This paper proposes a trojan attack mitigation method. The paper first observe that the the neural representation of clean and triggered samples are different in distribution. The proposed mitigation method first detects whether a model is attacked or not based on existing reverse engineering methods. For models that are considered attacked, the mitigation method minimizes the divergence between activation distributions of a clean input and a triggered input.



**Summary Of The Review:**

Overall, the idea is not quite novel. The solution involves multiple steps and heavily depends on existing attack detection methods. The mitigation-after-detection scheme is not very well justified.

---

> ### Author Response · Authors · 2022-11-14
> **Clarification for novelty and motivation**
>
> We thank the reviewers for providing valuable feedback to our work. We will improve the paper based on the reviews. Below we clarify misunderstandings and address questions for the reviewer.
> 1. Thanks for the suggestions on writing/presentation. We will improve it and add a pseudocode for our method.
> 2. Our work is not trivial but is novel: 1) Although one may think that it is not surprising that the distribution of backdoor-trigger instances is different from that of the clean instances in the internal layers (not only the penultimate layer), we are the *first* to theoretically prove the monotonicity between distribution divergence and source class inference accuracy (SIA) for backdoor-trigger instances. Besides, it is not easy to “pull the distributions together”. Please see the challenges and our proposed method in in Sec. 4.2. 2) Our another contribution is, based on the theoretical results, to mitigate the backdoor attacks in a post-training scenario by *solely* aligning the distributions, without touching the model parameters. As far as we know, we are the *first* to do this. We would appreciate it if you can provide references for similar ideas.  3) Also, tuning the model parameters seems to be the natural way to mitigate backdoor attacks, such as the state-of-the-art (SOTA) methods that we compared with in Sec. 5.2.  However, we did the *opposite*.  We demonstrated that some tuning based methods also correct the distribution alteration, but the performance (on both clean and backdoor-trigger instances) drops due to tuning many parameters using (insufficient) data. Moreover, parameter tuning requires much more computation than our method. Honestly, did you expect backdoor attacks can be well-mitigated without parameter tuning before reading our paper?
> 3. We agree that our mitigation method requires successful detection of the backdoor attack. However, we don’t agree that our method is not robust or cannot generalize well. 1) We think backdoor detection should be a standard protocol before mitigation, as we will elaborate in detail in the next clarification. 2) In fact, current backdoor detection approaches are effective against most trigger types and attack settings (as our repeated experiments in Table 1 in Sec. 5.2 show). It’s true that attacks are evolving, with stronger assumptions for the attacker. These attackers with “superpowers” may defeat a detector designed for practical attack scenarios. The other SOTA mitigation methods may not work on these (possibly future) attacks, either. Moreover, we used some classical detection methods in our experiments since they are effective against many attacks. Our method can be easily adapted to cooperate with other more powerful detectors to mitigate new types of attacks, which shows the generalization power of our method. 3) There are only two engineering components involved in our method – one for detection and the other for mitigation. For the detector, we chose widely applied detection methods like Neural Cleanse, which is generally acknowledged as an easy-to-tune method. For the mitigation part, it only involves three hyper-parameters – the bin size, the temperature, and the type of loss function (as we introduced in Sec. 4.2). In table 4 and table 10, we showed that our method is not sensitive to the choice of hyper parameters. Thus, our method is not hard to tune.
> 4. Thanks for pointing this out. We will justify the mitigation-after-detection scenario in our paper. First, detection and mitigation are two distinct problems.  A detection method infers the possible target class(es) (and reverse-engineers the associated backdoor trigger(s)). For an input that is classified as the target class, it still cannot tell if it is a backdoor-trigger sample or a clean target class sample (as discussed in Sec. 2). On the other hand, mitigation of the attacked model has the potential to correctly classify backdoor-trigger instances, just as for clean instances. That’s why many backdoor mitigation methods have been proposed, such as the methods proposed by Liu et al. (2018a); Wu & Wang (2021); Guan et al. (2022); Zheng et al. (2022)); Li et al. (2021c); Xia et al. (2022); and Zeng et al. (2022). Second, given that “mitigation” can in fact degrade accuracy (as we discussed in point 2 and in Sec. 5.2 in our paper), it should only be performed if an attack is detected – if there is no attack, “mitigation” may harm the accuracy of the classifier and is a waste of computation. That is also the reason that we do not agree with “a convincingly good mitigation method should work despite whether the model is attacked”. Third, mitigation-after-detection is analogous to repairing software after a virus has been detected and to curing a patient who has tested positive for covid.

---

> > ### Comment · Reviewer_2vpF · 2022-12-03
> > **Post rebuttal response**
> >
> > I thank the authors for responding to my initial review. I am still not fully convinced of the novelty mainly because the distributional shift of neural activation is not really surprising (as the authors agreed). I do agree that pulling distribution together will make the method stronger as it avoids explicit weight tuning. But I am not sure that makes the method fully robust as the method still depends on neuron activation and thus relies on the model architecture.
> >
> > My biggest issue is still the mitigation-after-detection design. I do not agree that detection is a matured technique despite many published papers. Indeed a common understanding is attacking is generally easier than detection. Therefore depending on given detection methods as the first step is not really a satisfying solution to me. I strongly disagree that mitigation will degrade clean sample accuracy even though I do not have hard evidence.
> >
> > Given these thoughts I still hold my original score.

---

### Decision · Program_Chairs · 2023-01-20

**Decision:**

Reject

**Justification For Why Not Higher Score:**

See above

**Justification For Why Not Lower Score:**

N/A

**Metareview: Summary, Strengths And Weaknesses:**

The work proposed a backdoor mitigation method by correcting the activation distribution based on a reverse-engineered triggers.

The reviewers indicated several important concerns about the motivation, the novelty, writing, experiments. Most of them are not well addressed after the rebuttal.